# Regulatory coiled-coil domains promote head-to-head assemblies of AAA+ chaperones essential for tunable activity control

Marta Carroni[1†]*, Kamila B Franke[2†], Michael Maurer[2], Jasmin Jäger[2], Ingo Hantke[3], Felix Gloge[4], Daniela Linder[2], Sebastian Gremer[2], Kürşad Turgay[3], Bernd Bukau[2]*, Axel Mogk[2]*

[1]Swedish Cryo-EM Facility, Science for Life Laboratory Stockholm University, Solna, Sweden; [2]DKFZ-ZMBH Alliance, Center for Molecular Biology of the University of Heidelberg (ZMBH) and German Cancer Research Center (DKFZ), Heidelberg, Germany; [3]Institute for Microbiology, Leibniz Universität Hannover, Hannover, Germany; [4]Wyatt Technology Europe, Dernbach, Germany

**\*For correspondence:**
marta.carroni@scilifelab.se (MC);
bukau@zmbh.uni-heidelberg.de (BB);
a.mogk@zmbh.uni-heidelberg.de (AM)

[†]These authors contributed equally to this work

**Competing interests:** The authors declare that no competing interests exist.

**Abstract** Ring-forming AAA+ chaperones exert ATP-fueled substrate unfolding by threading through a central pore. This activity is potentially harmful requiring mechanisms for tight repression and substrate-specific activation. The AAA+ chaperone ClpC with the peptidase ClpP forms a bacterial protease essential to virulence and stress resistance. The adaptor MecA activates ClpC by targeting substrates and stimulating ClpC ATPase activity. We show how ClpC is repressed in its ground state by determining ClpC cryo-EM structures with and without MecA. ClpC forms large two-helical assemblies that associate via head-to-head contacts between coiled-coil middle domains (MDs). MecA converts this resting state to an active planar ring structure by binding to MD interaction sites. Loss of ClpC repression in MD mutants causes constitutive activation and severe cellular toxicity. These findings unravel an unexpected regulatory concept executed by coiled-coil MDs to tightly control AAA+ chaperone activity.
DOI: https://doi.org/10.7554/eLife.30120.001

## Introduction

AAA+ (ATPase associated with a variety of cellular activities) proteins control a multitude of essential cellular activities by converting the energy derived from ATP hydrolysis into a mechanical force to remodel bound substrates (*Hanson and Whiteheart, 2005*). They are key players in protein quality control by targeting misfolded and aggregated proteins to degrading and refolding pathways. ClpB/Hsp104 reactivates aggregated proteins in concert with a cognate Hsp70 system (*Doyle et al., 2013*; *Mogk et al., 2015*). Other AAA+ proteins (e.g. ClpX, Rpt1-6) associate with peptidases (e.g. ClpP, 20S proteasome) to form AAA+ proteases, feeding protein substrates into associated proteolytic chambers for degradation (*Collins and Goldberg, 2017*; *Baker and Sauer, 2012*). The unfolding activity of AAA+ proteins can, however, also be deleterious to cells, in particular if linked to protein degradation, and therefore needs to be tightly controlled. Accordingly, loss of the control mechanisms of AAA+ protein activity in mutant proteins can lead to cell death (*Oguchi et al., 2012*; *Schirmer et al., 2004*; *Lipińska et al., 2013*). Controlling substrate selectivity and AAA+ protein activity is therefore crucial to prevent deleterious activities. This task is frequently executed by adapter proteins that select substrates but can also regulate the ATPase activity of AAA+ proteins and couple substrate delivery to ATPase activation (*Oguchi et al., 2012*; *Schlothauer et al., 2003*).

**eLife digest** If a protein does not fold into the correct shape, it may be unable to act correctly and can harm cells. As a result, cells contain biological machines that refold or break down misfolded proteins. ATP-dependent AAA+ proteases are an example of such machines. Their activity needs to be tightly controlled because breaking down the wrong proteins can also harm cells.

ATP-dependent AAA+ proteases form ring-shaped assemblies that are composed of AAA+ proteins and an associated peptidase. In bacteria, the AAA+ protein called ClpC can be crucial for resisting stress and infecting host cells. Adapter proteins help to activate ClpC by binding to extra domains that are fused to or inserted into the protein. This method of activation also requires repressing elements that ensure that the activity of ClpC remains low when the adapter proteins are not present. It was not known how this repression works.

Carroni, Franke et al. have now used a technique called cryo-electron microscopy to study the structures of repressed and adapter-activated ClpC from pathogenic bacteria called *Staphylococcus aureus*. In the repressed state, 10 molecules of ClpC interact to form two assemblies that interact via regions called middle domains. The middle domains have a "coiled coil" structure, and they interact via their ends in a head-to-head manner. In the repressed state ClpC cannot interact with its partner peptidase and the shape of the assembly shields the sites where adapter proteins can bind. This renders ClpC inactive.

Carroni, Franke et al. also studied ClpC proteins that had mutations to the middle domain that prevented the repressed state from forming. The mutant proteins remain in a constantly active state that is highly toxic to bacteria.

Bacteria are increasingly evolving to resist the effects of the antibiotics commonly used to treat infections. This is a severe problem, and we need to develop new antibiotic drugs that will kill these bacteria. AAA+ proteases have been identified as possible targets for new antibacterial drugs. The results presented by Carroni, Franke et al. suggest that disrupting the repressing activity of ClpC middle domains could be an effective way for such drugs to work.

DOI: https://doi.org/10.7554/eLife.30120.002

Activity control and adaptor action requires the ATPase activity to be repressed in the ground state, which is key to AAA+ chaperone mode of action. Repression can be achieved by regulatory coiled-coil domains inserted into an AAA module. In the ClpB/Hsp104 disaggregase a long coiled-coil middle domain (MD), consisting of two wings, is forming a repressing belt around the AAA ring to reduce ATPase activity (*Carroni et al., 2014*; *Heuck et al., 2016*). Adjacent MDs bind to each other by head-to-tail interactions keeping the regulatory domains in place. ATPase repression is relieved by MD dissociation and binding to Hsp70 adaptors that prevent reassociation of MDs with the ClpB/Hsp104 ring (*Oguchi et al., 2012*; *Rosenzweig et al., 2013*; *Lee et al., 2013*).

The bacterial AAA+ chaperone ClpC associates with the peptidase ClpP to form a central proteolytic machinery of Gram-positive bacteria. The ClpC-ClpP machinery acts in regulatory and general proteolysis, controlling multiple cellular pathways and differentiation processes and is crucial for bacterial stress resistance and virulence (*Turgay et al., 1997*; *Frees et al., 2014*; *Trentini et al., 2016*; *Capestany et al., 2008*; *Msadek et al., 1994*; *Krüger et al., 1994*; *Krüger et al., 2000*; *Lourdault et al., 2011*). ClpC activity crucially relies on cooperation with adaptor proteins including MecA, that target specific substrates while concurrently stimulating ClpC ATPase activity (*Schlothauer et al., 2003*; *Turgay et al., 1997*; *Mei et al., 2009*; *Turgay et al., 1998*). MecA binds to N-terminal and middle domains of each ClpC subunit forming a separate layer on top of the ClpC AAA ring (*Wang et al., 2011*). How ClpC is kept inactive in adaptor absence, and how the adaptor activates the ATPase is largely unknown. Furthermore, ClpC harbors a coiled-coil MD consisting only of a single wing, as opposed to the two-wing MD of ClpB/Hsp104. A potential regulatory function of the ClpC MD has not been investigated, but its smaller size as compared to the ClpB/Hsp104 MD implies it must act differently if involved in ClpC activity control. Understanding ClpC regulation is particularly relevant as AAA+ protease machines including ClpC have attracted considerable attention as targets for antibacterial action in recent years (*Brötz-Oesterhelt and Sass, 2014*). Overruling

AAA+ protease control by small molecules can lead to constitutive uncontrolled and toxic activation as best exemplified by acyldepsipeptide antibiotics of the ADEP class targeting the ClpP peptidase. ADEP-activated ClpP causes aberrant protein degradation and even allows for eradication of *Staphylococcus aureus* persister cells (*Conlon et al., 2013*; *Brötz-Oesterhelt et al., 2005*; *Kirstein et al., 2009*). Understanding ClpC activity control therefore might open new avenues for antibiotics development.

Here, we report on an unexpected mode of AAA+ chaperone control involving transition between an inactive resting state and a functional hexamer as revealed by determining the cryoEM-structures of *S. aureus* ClpC in absence and presence of MecA. The ClpC resting state is composed of two helical ClpC assemblies stabilized by head-to-head MD interactions. MecA prevents MD interactions and thereby converts ClpC into a canonical and active hexamer.

## Results

### The ClpC M-domain represses ClpC activity

To study the function of the M-domain (MD) in ClpC activity control we first purified *S. aureus* ClpC/ClpP and demonstrated functionality by determining high-proteolytic activity in presence of the adaptor MecA (*Figure 1*). Next, we created a series of ClpC MD variants by mutating conserved residues not involved in coiled-coil structure formation (*Figure 1—figure supplement 1A*). Additionally, we replaced the entire MD (N411-K457) by a di-glycine linker, allowing MD deletion without interfering with folding of the AAA-1 domain. Proteolytic activities of MD mutants were determined using Fluorescein-labeled casein (FITC-casein) as constitutively misfolded model substrate in absence and presence of MecA (*Figure 1A/B*). ClpC wild type (WT) together with ClpP exhibited only a low proteolytic activity in absence of MecA and FITC-casein degradation rates were 20-fold increased upon adaptor addition. In contrast, most MD mutants enabled for adaptor-independent FITC-casein proteolysis to varying degrees. ClpC-F436A, ClpC-R443A and ClpC-D444A showed highest activities with degradation rates close to those determined for ClpC WT plus MecA (*Figure 1A/B*). Similarly, MD deletion strongly increased ClpC activity, indicating that the single point mutants reflect a loss of M-domain function. MecA presence still stimulated FITC-casein degradation by ClpC MD mutants except F436A and ΔM, consistent with the crucial function of F436 in MecA binding (*Figure 1A*) (*Wang et al., 2011*). To analyze whether M-domain mutants cause full activation of ClpC, we compared FITC-casein degradation rates of ClpC-F436A and ClpC/MecA under saturating conditions (*Figure 1—figure supplement 1B/C*). ClpC-F436A degraded FITC-casein with similar efficiencies as ClpC/MecA at all substrate concentrations tested and reached identical $v_{max}$. ClpC-R443A and ClpC-ΔM also degraded FITC-casein at saturating concentrations like MecA-activiated ClpC, underlining complete activation of ClpC upon M-domain mutation (*Figure 1—figure supplement 1C*) Notably, we observed minor FITC-casein degradation by ClpC at higher substrate concentrations and indicating partial ClpC activation without adapter.

FITC-casein degradation by activated ClpC M-domain mutants (F436A, ΔM) required ATP hydrolysis and was not observed in presence of ATPγS (*Figure 1—figure supplement 1D*). Complete degradation of FITC-casein by ClpC-F436A was confirmed by SDS-PAGE (*Figure 1—figure supplement 2*), while ClpC WT required MecA to exhibit proteolytic activity. Here, we also noticed degradation of MecA degradation once FITC-casein was digested, in agreement with former findings for the *B. subtilis* ClpC/MecA system (*Schlothauer et al., 2003*; *Mei et al., 2009*; *Turgay et al., 1998*). We infer that ClpC MD mutants exhibit high, adaptor-independent proteolytic activities, qualifying the M-domain as a negative regulatory element.

To further investigate a repressing function of the ClpC MD we determined GFP-SsrA degradation activities of selected ClpC MD mutants exhibiting highest FITC-casein degradation activities (F436A, R443A, ΔM). GFP-SsrA harbors the 11-meric SsrA tag, which is recognized by AAA+ chaperrones pore sites (*Hinnerwisch et al., 2005*; *Piszczek et al., 2005*; *Siddiqui et al., 2004*). GFP-SsrA degradation requires application of high unfolding force in contrast to FITC-casein, which is constitutively unfolded. ClpC WT/ClpP efficiently degraded GFP-SsrA in a MecA-dependent manner (*Figure 1C*). Surprisingly, ClpC MD mutants did hardly exhibit autonomous degradation of GFP-SsrA in contrast to FITC-casein. ClpC-R443A was partially stimulated upon MecA addition demonstrating that the M-domain mutant can process GFP-SsrA in principle (*Figure 1—figure supplement 3A*).

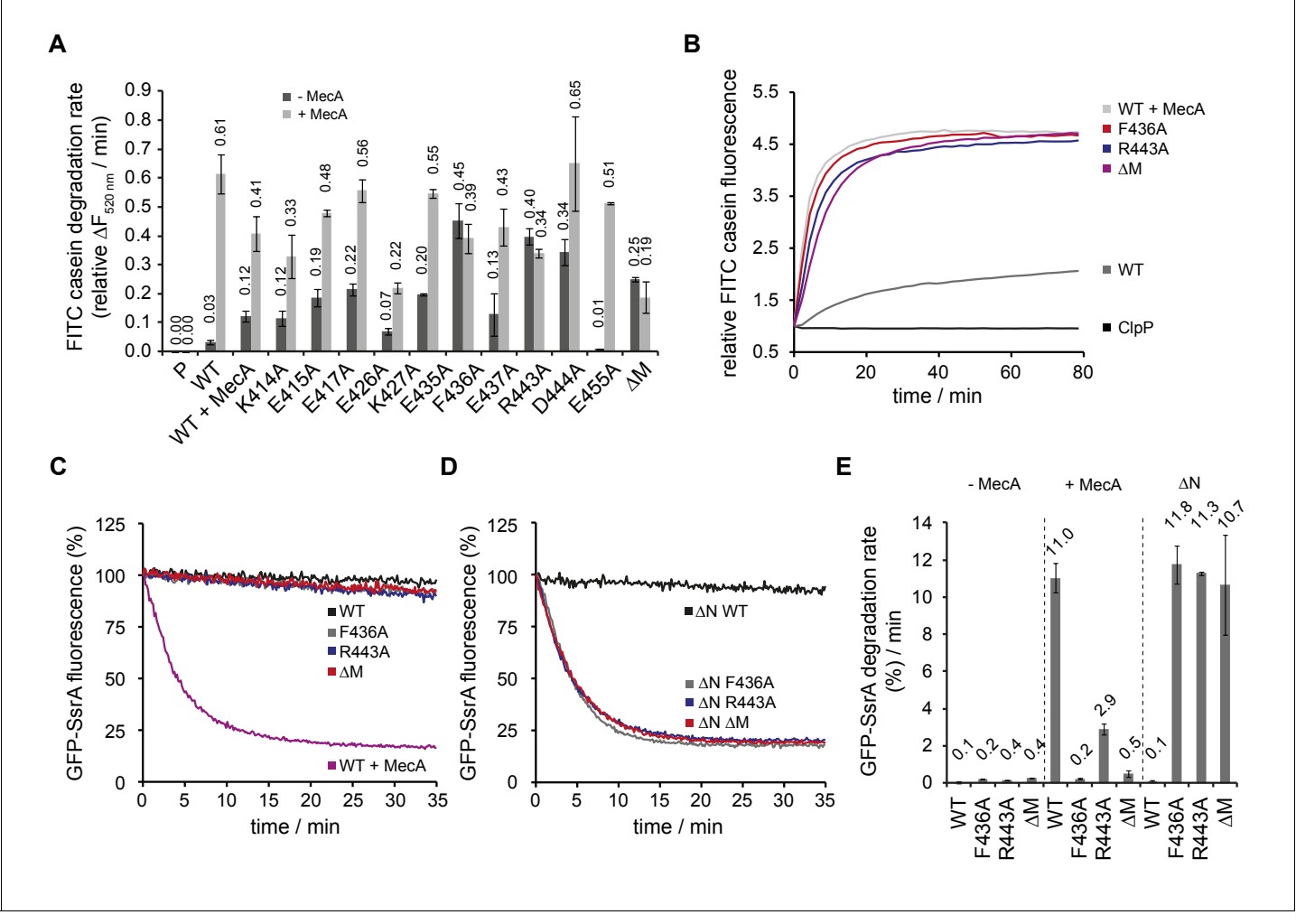

**Figure 1.** ClpC MD mutants exhibit adaptor-independent proteolytic activity. (A/B) FITC-casein degradation was monitored in the presence of ClpP (P) only, or in presence of ClpC wild type and indicated MD mutants with or without MecA. Degradation rates were determined from the initial linear increase of FITC fluorescence. Initial FITC-casein fluorescence was set as one and relative changes in fluorescence were recorded. (C–E) GFP-SsrA degradation was monitored in the presence of ClpP and indicated ClpC variants. Deletion of the N-terminal domain (ΔN) unleashes high proteolyic activity of MD mutants. GFP-SsrA degradation rates were determined from the initial linear decrease of GFP-SsrA fluorescence.

DOI: https://doi.org/10.7554/eLife.30120.003

The following figure supplements are available for figure 1:

**Figure supplement 1.** Analysis of ClpC MD mutants.
DOI: https://doi.org/10.7554/eLife.30120.004

**Figure supplement 2.** Degradation of FITC-casein by the ClpC-F436A MD mutant.
DOI: https://doi.org/10.7554/eLife.30120.005

**Figure supplement 3.** Degradation of GFP-SsrA by ClpC MD mutants.
DOI: https://doi.org/10.7554/eLife.30120.006

We speculated that differences in GFP-SsrA and FITC-casein binding modes might be the cause of the different degradation activities of ClpC MD mutants. Hsp100 N-domains contribute to casein binding (*Beinker et al., 2002*; *Rosenzweig et al., 2015*) while their position on top of the AAA-1 ring and central pore site could impede GFP-SsrA binding. We therefore determined GFP-SsrA degradation activities of N-domain deleted ΔN-ClpC and respective MD mutants (*Figure 1D*). GFP-SsrA remained stable in presence of ΔN-ClpC/ClpP, however, the substrate was rapidly degraded by ΔN-ClpC-F436A, ΔN-ClpC-R443A and ΔN-ClpC-ΔM (+ClpP) and degradation rates were identical to those determined for ClpC WT/ClpP with MecA (*Figure 1E*). We did not test for GFP-SsrA

degradation by ΔN-ClpC/ClpP in presence of MecA, as the N-domain is essential for MecA binding (*Wang et al., 2011*; *Kirstein et al., 2006*; *Persuh et al., 1999*).

By determining GFP-SsrA degradation rates under saturating conditions we confirmed that ΔN-ClpC-F436A is as active as MecA-activated ClpC and degraded the substrate with similar or even faster kinetics (*Figure 1—figure supplement 3B*). Similar results were obtained for ΔN-ClpC-R443A and ΔN-ClpC-ΔM when determining degradation rates in presence of 30-fold GFP-SsrA excess (*Figure 1—figure supplement 3C*).

GFP-SsrA degradation by activated ΔN-ClpC M-domain mutants relied on ATP hydrolysis and remained specific as the substrate variant GFP-SsrA-DD was not degraded (*Figure 1—figure supplement 3D*). Here, the two C-terminal alanine residues of the SsrA tag are replaced by aspartate residues, obstructing binding to the AAA+ chaperone pore site (*Flynn et al., 2001*). This documents that MD mutations boost ClpC unfolding activity in absence of adaptor without altering general substrate specificity.

## ClpC M-domain mutants exhibit increased basal ATPase activities

*B. subtilis* ClpC activation by adaptors involves strong stimulation of ClpC ATPase activity (*Turgay et al., 1997*; *Mei et al., 2009*; *Persuh et al., 1999*), which we confirm here for *S. aureus* ClpC and MecA (*Figure 2A*). ClpC WT exhibits a very low basal ATPase activity (0.4 ATP/min/monomer), rationalizing its poor standalone degradation activity. Notably, all activated MD mutants (F436A, R443A, ΔM) exhibited strongly increased basal ATPase activities (3.3–5.4 ATP/min/monomer) that were further increased by substrate casein (1.64–2.47-fold stimulation) in contrast to ClpC WT (*Figure 2A/B*). In presence of MecA, a minor stimulation (1,21-fold) of ClpC ATPase activity by casein was determined. In this setup lower MecA concentrations (0.2 μM) were used to minimize competition between casein and MecA as both can act as ClpC substrate. Casein-stimulated ClpC MD mutants reached 33% of total ATPase activity determined for MecA-activated ClpC WT. N-domain deletion also increased basal ATPase activity of ΔN-ClpC. Combining ΔN-ClpC with M-domain mutants had an additive stimulatory effect on ATPase activities, suggesting that the underlying mechanisms are distinct from one another (*Figure 2A*). We infer increased basal ATPase activities of ClpC MD mutants (in context of fullength ClpC and ΔN-ClpC) can explain adaptor-

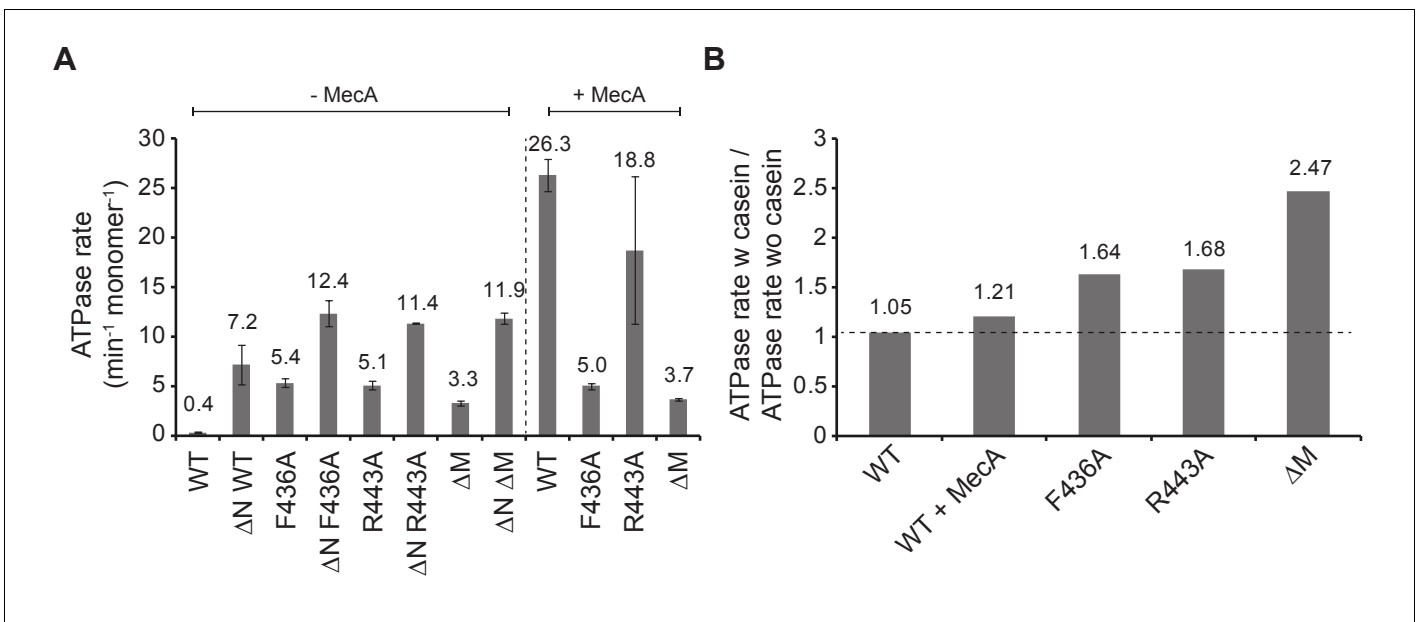

**Figure 2.** ClpC MD mutants exhibit increased basal ATPase activity that can be stimulated by substrate. (**A**) ATPase activities of ClpC wild type (WT) and indicated deletion variants (ΔN, ΔM) and MD mutants were determined in absence and presence of MecA. (**B**) ATPase activities of ClpC wild type (WT, ±MecA) and indicated MD mutants were determined in absence and presence of substrate casein. ATPase activities determined without casein were set as one and the relative increase of ATP hydrolysis in presence of casein was determined (stimulation factor).
DOI: https://doi.org/10.7554/eLife.30120.007

independent substrate degradation. However, ATPase activation alone is not sufficient to explain ClpC activation as ΔN-ClpC did not allow degradation of GFP-SsrA despite having increased basal ATPase activity. This indicates specific consequences of MD mutations on ClpC conformation and activity.

## Head-to-head interactions of M-domains mediate formation of an inactive ClpC resting state

To understand the structural basis underlying the control of ClpC activity by the MD, we determined the cryo-EM structures of *S. aureus* ClpC with and without its activator MecA in the presence of ATPγS. The structure of ClpC on its own was solved at 8.4 Å resolution using ~90.000 collected particles (*Figure 3—figure supplement 1*). Surprisingly, raw images and subsequent 2D classification revealed immediately that ClpC assumes a conformation different from the canonical hexameric arrangement of AAA+ proteins (*Figure 3—figure supplement 1A/B*). Attempts of 3D classification and refinement using the existing ClpC-MecA structures (*Wang et al., 2011*; *Liu et al., 2013*) failed to give a refined map, indicating that on its own, ClpC assumes a substantially different conformation. Indeed, reconstructions revealed that ClpC without MecA assembles into an oligomeric assembly made of two open spirals that interact via head-to-head contacts mainly mediated by the MDs (*Figure 3A/B*, *Figure 3—figure supplement 1*). Continuous spiraling of AAA+ proteins is often observed in X-ray structures (*Carroni et al., 2014*; *Heuck et al., 2016*; *Lee et al., 2003*; *Guo et al., 2002*) and ClpC half spirals are similar to these crystal packings. More specifically, ClpC half spirals have inter-subunit interfaces similar to the crystal structure of Hsp104 (PDB code: 5d4w) (*Heuck et al., 2016*) and the EM map of Hsp104 in complex with ADP (PDB code: 5vy8) (*Gates et al., 2017*). ClpC did not form a half spiral but a double spiral that is open on one side (*Figure 3A/B*, *Figure 3—figure supplement 1A/B*, *Figure 3—video 1*) resulting in a cradle-like molecule with the peripheral subunits more mobile than the core ones, as shown by lower local resolution (*Figure 3—figure supplement 2A*). Additionally, high-threshold noise on the open part of the spiral indicates higher dynamics of this region, suggesting exchange of subunits. Accordingly, for peripheral ClpC subunits there is not sufficient density to account for all ClpC domains (*Figure 3—figure supplement 2B*). Local focused 3D classification of these peripheral regions failed to give well-defined density further suggesting high-mobility.

The overall subdomain organization within the ClpC protomer is similar to that of the ClpC-MecA crystal structures, however domain positions and interactions are different. The AAA+ domains are staggered with a rise of ~20 Å per subunit (*Figure 3B*). N-domains domains are packed between MDs and are displaced so that they lie on top of the adjacent small AAA1 subdomain (*Figure 3B/C*). The MDs coiled-coils constitute the backbone of the spiral and mediate the spiral head-to-head contacts, which involve residues F436, R443 and D444 providing a structural rationale for the activated states of respective MD mutants (*Figure 3D*). The MD-MD head-to-head interaction stops the formation of virtually infinite spirals as in the Hsp104 crystal arrangements allowing the formation of a compact, repressed ClpC reservoir.

We envision the helical ClpC assembly as a dynamic inactive resting state, which will interfere with substrate binding, association of the ClpP peptidase and interaction with adaptor proteins.

The *S. aureus* ClpC-MecA cryo-EM map was reconstructed at 11 Å resolution with ~26,000 particles, (*Figure 3—figure supplement 3A–D*) and shows the classical hexameric assembly previously described (*Wang et al., 2011*; *Liu et al., 2013*), with MecA interacting with both the MD and the N-domain of ClpC (*Figure 3E*). Additional extra density caps the hexamer and accounts for the N-terminal domain of MecA, whose structure is unknown. From the resting to the MecA-bound state the N-domains undergo a 45° rotation that repositions the MecA-binding loop region from being blocked by the MD of the neighbouring subunit to be available and engaged in MecA binding (*Figure 3—figure supplement 3E/F*). Additionally, binding of MecA to MDs breaks head-to-head MD contacts and is therefore expected to prevent formation of the ClpC resting state.

Taken together, our structures of ClpC alone or in complex with MecA show a dramatic reorganization from a helical resting state to a planar canonical AAA+ hexamer, explaining ClpC activation by the adaptor MecA. Intermolecular head-to-head contacts of MDs form the backbone of the ClpC resting state, qualifying the MD as crucial negative regulatory element consistent with MD mutant characterization.

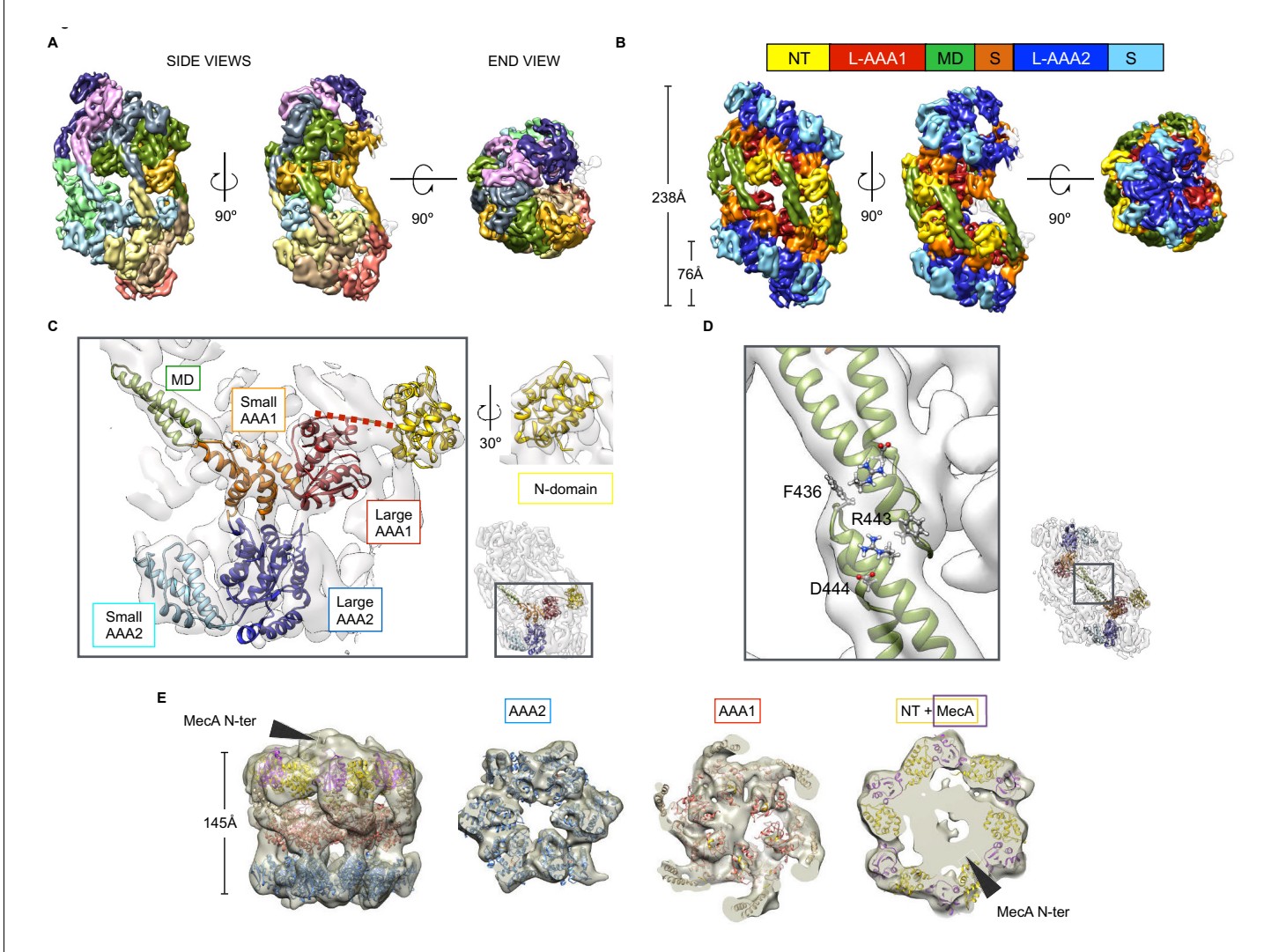

**Figure 3.** Cryo-EM structures of *S. aureus* ClpC-ATPγS with and without MecA. (**A**) Overview of the oligomeric ClpC resting state 3D map coloured by subunits. (**B**) ClpC density map coloured by domains. Head-to-head interactions between MDs (green) of each helical assembly are key contacts stabilizing the ClpC resting state. ClpC domain organization is given, including an N-terminal domain (NT, yellow), AAA-1 large subdomain (L-AAA1, red), coiled-coil M-domain (MD, green), AAA-1 small subdomain (S, orange), AAA-2 large subdomain (L-AAA2, blue) and AAA-2 small subdomain (S, cyan). (**C**) Details of fitting for each ClpC subdomain. The small inset shows the positions relative to the whole molecule. (**D**) Zoomed view into MD-MD contacts highlighting conserved MD residues involved in interactions. The small inset shows the positions relative to the whole molecule. (**E**) Structure of the ClpC-MecA complex. Fitted atomic model coloured by ClpC domains as in b and MecA C-terminal domains (purple). Density for MecA N-terminal domains is indicated.

DOI: https://doi.org/10.7554/eLife.30120.008

The following video and figure supplements are available for figure 3:

**Figure supplement 1.** Cryo-EM structure of *S.*
DOI: https://doi.org/10.7554/eLife.30120.009

**Figure supplement 2.** (A) Local-resolution map of ClpC-ATPγS complex.
DOI: https://doi.org/10.7554/eLife.30120.010

**Figure supplement 3.** Cryo-EM structure of *S. aureus* ClpC-MecA-ATPγS complex.
DOI: https://doi.org/10.7554/eLife.30120.011

**Figure 3—video 1.** Structural organization of the ClpC resting state.
DOI: https://doi.org/10.7554/eLife.30120.012

## MecA abrogates head-to-head M-domain interactions

To demonstrate direct head-to-head contacts of ClpC MDs we introduced cysteine residues at the tip of the MD (E435C, E437C) to probe for site-specific disulfide crosslinking. These mutations were introduced into ClpC-C311T to avoid interference of the endogenous Cys311 residue. ClpC-C311T/E435C and ClpC-C311T/E437C were active in MecA-dependent protein degradation under reducing conditions (*Figure 4—figure supplement 1A*). Formation of crosslink products under oxidizing conditions that were fully reverted by addition of reducing agent was observed for ClpC-C311T/E437C but not for ClpC-C311T/E435C and the ClpC-C311T control (*Figure 4—figure supplement 1B*). This documents specificity of disulfide crosslinking and agrees well with the ClpC resting state model, showing E437 residues of two interacting M-domains are facing one another while E435 residues are oriented in opposite directions (*Figure 4—figure supplement 1B*). E437C disulfide crosslinking was most efficient in absence of nucleotide or presence of ADP and ATP, and less efficient in presence of ATPγS (*Figure 4—figure supplement 1C*). Importantly, ClpC-E437C crosslinking in presence of MecA was strongly reduced (*Figure 4A*), consistent with MecA binding to MDs preventing head-to-head MD interactions. Similarly, crosslinking efficiency was reduced for ClpC-F436A/E437C, indicating a crucial contribution of F436 to intermolecular MD contacts (*Figure 4B*), consistent with the ClpC WT cryo-EM structure.

## Obstructing head-to-head M-domain contacts allows for ClpC hexamer formation

To provide biochemical support for the formation of a large, inactive ClpC resting state we employed chemical crosslinking and size exclusion chromatography. We used the ATPase-deficient ClpC-E280A/E618A variant (referred to as ClpC-DWB), harboring mutated Walker B motifs in both AAA+ domains allowing for ATP binding but not hydrolysis, facilitating analysis of adaptor or substrate impact on ClpC assembly. We first determined sizes of ClpC assemblies by glutaraldehyde crosslinking (*Figure 5A*). ClpC-DWB was crosslinked to very large assemblies that were just entering the separating gel in SDS-PAGE in absence and presence of ATP (*Figure 5A*). Addition of MecA allowed for formation of a smaller high molecular weight complex similar in size to crosslinked ClpB hexamers that were used as reference (*Figure 5A*). Presence of MecA in the crosslinked ClpC-DWB complexes was confirmed by western-blot analysis (*Figure 5—figure supplement 1*). We still observed crosslinking of ClpC to high molecular weight products in presence of MecA. These might stem from transient interactions between ClpC/MecA hexamers via MecA-MecA interactions as MecA can target itself for degradation. ClpC-F436A-DWB stayed monomeric in absence of nucleotide, indicating that large assemblies observed for ClpC-DWB rely entirely on MD contacts (*Figure 5A*). ClpC-F436A-DWB crosslinking in presence of ATP caused formation of defined ClpC-F436A-DWB complexes that were similar in size to crosslinked ClpB hexamers. These findings confirm the predicted critical contribution of the MD to formation of a large resting state and the role of MecA in converting this assembly into a functional hexamer.

Since N-domains are packed between MDs in the resting state and lie on top of the adjacent AAA-1 domain we analyzed oligomerization of ΔN-ClpC and ΔN-ClpC-F436A by crosslinking (*Figure 5B*). In absence of nucleotide ΔN-ClpC stayed monomeric, suggesting that N-domain contacts to AAA-1 contribute to assembly formation. However, in presence of ATPγS, ΔN-ClpC was still crosslinked to high molecular weight complexes larger than ClpB hexamers indicating N-domains are not essential for resting state formation whereas nucleotide binding to AAA domains is crucial (*Figure 5B*). ΔN-ClpC-F436A (+ATPγS) was predominantly crosslinked to hexamers, demonstrating the unique and dominant role of MDs in controlling resting state formation.

We next analyzed sizes of ClpC complexes by size exclusion chromatography (*Figure 5C*, *Figure 5—figure supplement 2A*). In absence of ATP ClpC-DWB showed a broad elution profile, suggesting formation of variable assemblies ranging from monomers to hexamers. ATP addition caused formation of larger ClpC assemblies that eluted prior to ClpB hexamers and a 670 kDa standard protein, suggesting formation of ClpC complexes larger than hexamers (*Figure 5C*, *Figure 5—figure supplement 2A*). This was confirmed by static light scattering (SLS) measurements, revealing a molecular mass of ≈ 956 kDa corresponding to a decameric complex, consistent with cryoEM analysis (*Figure 5—figure supplement 2B*). Presence of MecA sharpened the ClpC elution profile and shifted ClpC-DWB fractions to later elution volumes right after the 670 kDa standard protein and

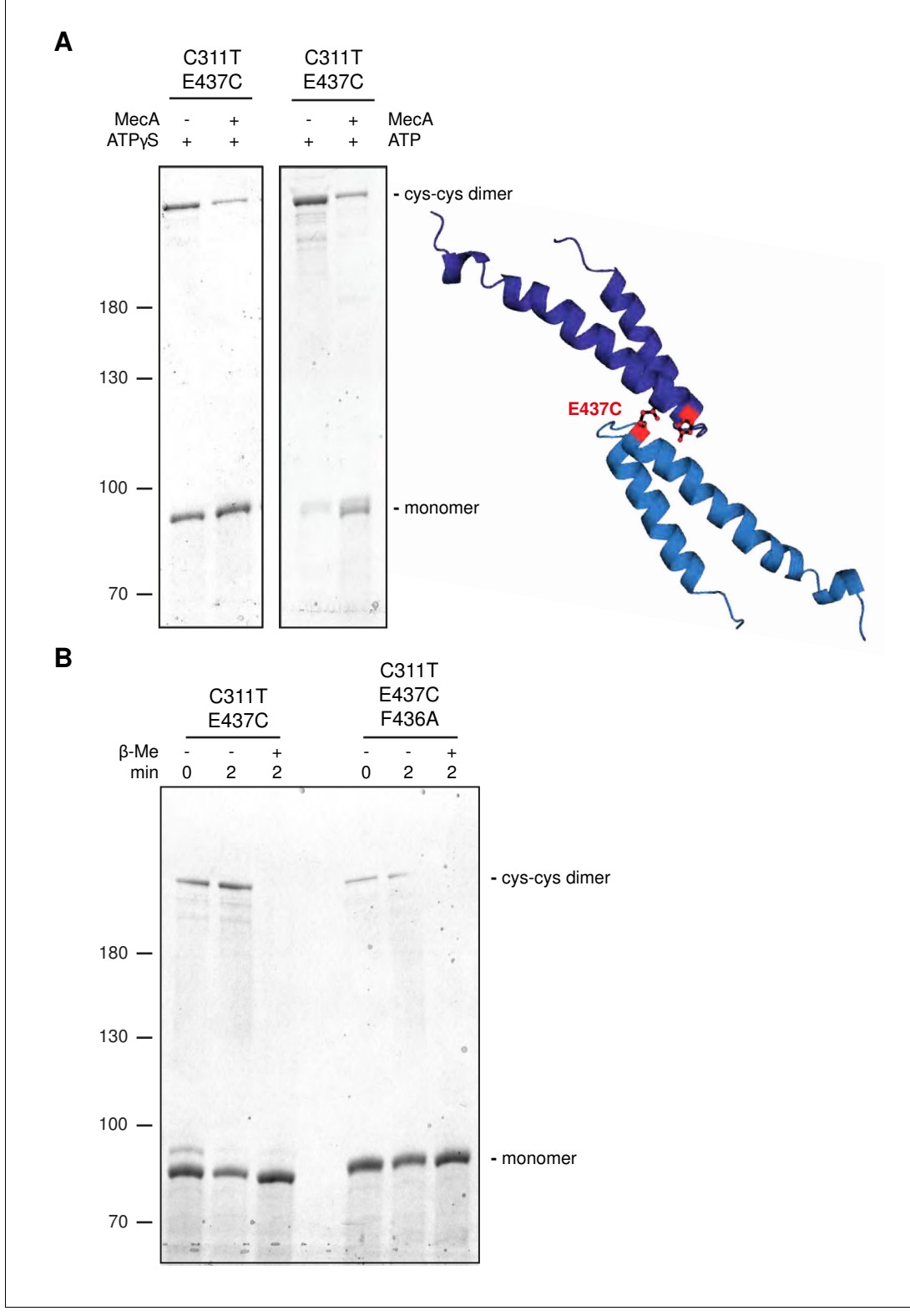

**Figure 4.** Disulfide crosslinking demonstrates MecA-sensitive head-to-head MD contacts. (**A**) Disulfide crosslinking of ClpC-C311T/E437C under oxidizing conditions was performed in presence of ATPγS or ATP without or with MecA and analyzed by subsequent non-reducing SDS-PAGE. A model of head-to-head interacting MDs is given and the position of E437 is indicated. (**b**) Disulfide crosslinking of ClpC-C311T/E437C and ClpC-C311T/F436A/
*Figure 4 continued on next page*

*Figure 4 continued*

E437C was performed in presence of ATPγS under oxidizing (+Cu(Phe₃)) and reducing (+ β-mercaptoethanol) conditions and analyzed by subsequent non-reducing SDS-PAGE.
DOI: https://doi.org/10.7554/eLife.30120.013

The following figure supplement is available for figure 4:

**Figure supplement 1.** Disulfide crosslinking of head-to-head interacting MDs.
DOI: https://doi.org/10.7554/eLife.30120.014

now overlapping with ClpB hexamers (*Figure 5C*, *Figure 5—figure supplement 2A*). Quantification of co-eluting MecA suggests the formation of a 1:1 ClpC:MecA complex, consistent with the binding stoichiometry determined for *B. subtilis* ClpC/MecA complexes and mass determination by SLS (767 kDa) (*Figure 5—figure supplement 2B*) (*Wang et al., 2011*; *Kirstein et al., 2006*). This indicates that MecA binding shifts ClpC from a large, non-hexameric assembly to a ClpC₆/MecA₆ complex. We next analyzed the elution profile of ClpC-DWB-F436A. In absence of ATP the MD mutant eluted as defined species right at the elution volume of a 158 kDa standard protein suggesting formation of monomers/dimers (*Figure 5C*). This indicates that the formation of larger assemblies noticed for ClpC WT (- ATP) depends on the M-domain, consistent with results from glutaraldehyde crosslinking (*Figure 5A*) Unexpectedly, ClpC-DWB-F436A showed a broad elution profile upon ATP addition. While the ClpC-DWB-F436A peak fraction eluted right after the 670 kDa standard, larger assemblies at earlier elution volumes were also present. We speculated that activated ClpC MD mutants might be capable of recognizing themselves as substrate resulting in formation of larger complexes and explaining the elution profile. Indeed, we observed autodegradation of ClpC-F436A, ClpC-R443A and ClpC-ΔM but not ClpC WT in presence of ClpP (*Figure 5—figure supplement 3*). To prevent self-recognition of ClpC-DWB-F436A we repeated the size exclusion analysis in presence of substrate casein excess (*Figure 5C*). Presence of casein sharpened the ClpC-DWB-F436A elution profile that was comparable to ClpC-DWB/MecA complexes and distinct from ClpC-DWB, suggesting hexamer formation, which was confirmed by mass determination by SLS (587 kDa) (*Figure 5C*, *Figure 5—figure supplement 2B*). In contrast, casein addition did not cause formation of smaller ClpC-DWB complexes (*Figure 5—figure supplement 2B*). To further prove ClpC-F436A hexamer formation, a small cryo-EM dataset of ClpC-F436A with casein and ATPγS was collected and analyzed via 2D classification. Classes indicate that the ClpC-F436A assembles in an hexamer similar to ClpC WT with MecA (*Figure 5D*).

Together our findings demonstrate that ClpC-DWB forms large, non-hexameric assemblies in a MD dependent manner, supporting the derived ClpC WT cryo-EM structure. This large ClpC-DWB assembly should neither allow for efficient substrate binding nor association with the ClpP peptidase. This prediction was confirmed by size exclusion chromatography showing poor interaction with ClpP and negligible binding to substrate FITC-casein (*Figure 5E*, *Figure 5—figure supplement 2C*). In contrast, efficient binding to FITC-casein and ClpP was observed upon addition of MecA and for ClpC-DWB-F436A (- MecA) (*Figure 5E*, *Figure 5—figure supplement 2C*). These findings were further confirmed by monitoring FITC-casein binding by anisotropy measurements (*Figure 5—figure supplement 2D/E*).

## M-domain activity control of ClpC is crucial for cellular viability

ClpC MD mutants allow for adaptor-independent and thus constitutive and uncontrolled ClpC activity. We wondered whether this loss of ClpC activity control has physiological consequences and co-expressed *S. aureus* ClpC WT, ΔN-ClpC and respective F436A mutants from an IPTG-regulated promoter together with *S. aureus* ClpP in *E. coli* cells. This strategy allowed us to only monitor potential toxic effects of ClpC M-domain mutants without interference by loss of endogenous functions of ClpC/MecA in *S. aureus* cells. Levels of ClpC variants were similar after 1 hr of IPTG-induced protein production (*Figure 6—figure supplement 1A*). In case of ClpC-F436A we noticed accumulation of a degradation product upon ClpP coexpression, as also observed in vitro (*Figure 5—figure supplement 3*), suggesting autoprocessing. We observed strong toxicity upon expression of ΔN-ClpC-F436A at all temperatures tested while ClpC-F436A expression became lethal at 37°C and 40°C (*Figure 6A*). Toxicity of ClpC MD mutants was much higher as compared to ClpC-WT and ΔN-ClpC,

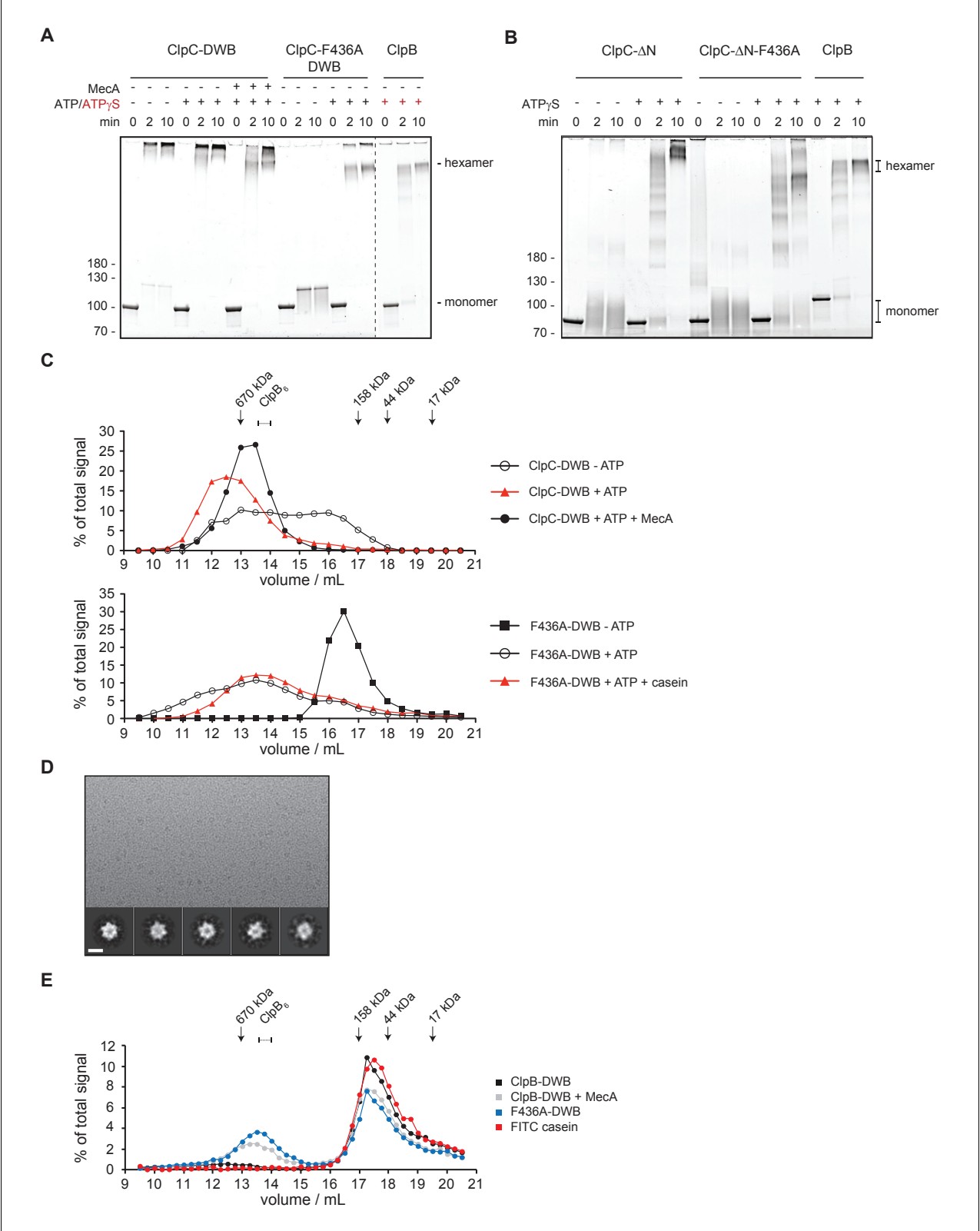

**Figure 5.** ClpC forms a large, inactive resting state that is sensitive to MecA and MD mutation. (**A**) Glutaraldehyde crosslinking of ClpC-E280A/E618A (ClpC-DWB) and a respective MD mutant variant (F436A) was performed in absence and presence of ATP without and with MecA as indicated. Crosslinking of *E. coli* ClpB in presence of ATPγS served as reference defining crosslinked hexameric assemblies. Crosslink products were analyzed by SDS-PAGE. (**B**) Glutaraldehyde crosslinking of ΔN-ClpC and ΔN-ClpC-F436A was performed in absence and presence of ATPγS as indicated.
*Figure 5 continued on next page*

*Figure 5 continued*

Crosslinking of *E. coli* ClpB in presence of ATPγS served as reference defining crosslinked hexameric assemblies. Crosslink products were analyzed by SDS-PAGE. (C) Oligomeric states of ClpC-DWB and ClpC-F436A-DWB were determined in absence and presence of ATP. Addition of MecA and casein is indicated. Elution fractions were analyzed by SDS-PAGE and quantified. Positions of peak fractions of a protein standard and ClpB-E279A/E678A hexamers (+ATP) are indicated. (D) Micrograph of ClpC-F436A sample in presence of casein (top). Examples of single particles are circled. 2D class averages of ClpC-F436A (bottom). Scale bar is 10 nm. (E) Binding of FITC-casein to ClpC-DWB and ClpC-F436A-DWB was analyzed in presence of ATP by size exclusion chromatography. FITC-casein fluorescence of elution fractions was quantified. Positions of peak fractions of a protein standard and ClpB-E279A/E678A hexamers (+ATP) are indicated.

DOI: https://doi.org/10.7554/eLife.30120.015

The following figure supplements are available for figure 5:

**Figure supplement 1.** MecA is crosslinked to a high molecular weight complex upon ClpC binding.

DOI: https://doi.org/10.7554/eLife.30120.016

**Figure supplement 2.** ClpC-ATP forms an inactive large resting state.

DOI: https://doi.org/10.7554/eLife.30120.017

**Figure supplement 3.** Autodegradation of ClpC MD mutants.

DOI: https://doi.org/10.7554/eLife.30120.018

suggesting an essential cellular need for ClpC repression by MDs. Furthermore, toxicity of ClpC MD mutants was dependent on coexpression of *S. aureus* ClpP, suggesting that uncontrolled protein degradation caused by constitutively activation results in cell death (*Figure 6—figure supplement 1B*). We also noticed toxic effects when co-expressing MecA, ClpC and ClpP in *E. coli* cells (*Figure 6—figure supplement 1C*), substantiating that ClpC-F436A toxicity is caused by its constitutive activation. Toxicity of MecA/ClpC/ClpP was somewhat lower as compared to e.g. ClpC-F436A, but eventually also restricted by strong MecA degradation (*Figure 6—figure supplement 1D*).

To further explore a cellular need for ClpC activity control we determined the physiological consequences of the same ClpC MD mutation in *B. subtilis* cells harboring a complete set of ClpC adapter proteins. Here, we deleted the chromosomal copy of the *clpC* gene (*clpC::tet*) and re-integrated either *clpC wt* or *clpC-F436A* at the *amyE*-locus under IPTG control. Expression of *clpC-F436A* but not *clpC wt* in presence of 100 μM IPTG was highly toxic at all temperatures (30–50°C) (*Figure 6B*). Levels of ClpC-wt and ClpC-F436A produced after IPTG addition in cells cultured in liquid medium were comparable, excluding differences in protein levels as reason for toxicity (*Figure 6—figure supplement 1E*). Importantly, growth of *clpC::tet* cells was not impaired, demonstrating that toxicity of ClpC-F436A reflects a gain-of-function phenotype and is not caused by loss of adaptor interaction (*Figure 6B*). Together these findings demonstrate an essential role of MD mediated activity control for cellular viability.

The Hsp100 family members ClpE and ClpL harbor a coiled-coil MD that is similar in size to the ClpC MD and also displays some sequence homology. Notably, F436 and R443, identified here as key MD residues in ClpC activity control, are largely conserved in ClpE and ClpL M-domains (*Figure 6C*). This suggests that the role of M-domains as crucial negative regulators of Hsp100 activity is conserved in other family members. To test for a conserved regulatory function we generated the *B. subtilis* ClpE-Y344A MD mutant corresponding to ClpC-F436A. *clpE wt* and *clpE-Y344A* copies were integrated at the *amyE*-locus in *B. subtilis* wild type cells under control of an IPTG-regulatable promoter. *B. subtilis* cells do hardly express endogenous *clpE* at non-stress conditions (*Gerth et al., 2004*; *Miethke et al., 2006*) thereby allowing mutant analysis without interference of ClpE WT copies. Expression of *clpE-Y344A* but not *clpE wt* in presence of 100 μM IPTG was highly toxic to *B. subtilis* cells at all temperatures tested (30–50°C) (*Figure 6D*). ClpE WT and ClpE-Y344A were produced to similar levels underscoring that toxicity is caused by deregulation of the ClpE MD mutant (*Figure 6—figure supplement 1F*). This suggests that ClpE MDs are also essential to downregulate ClpE activity preventing cellular toxicity.

## Discussion

In the presented work we established a new mechanism of activity control of ClpC, a central AAA+ chaperone widely distributed among Gram-positive bacteria. We show that coiled-coil MDs control ClpC activity in a unique manner by sequestering ClpC molecules in an inactive resting state.

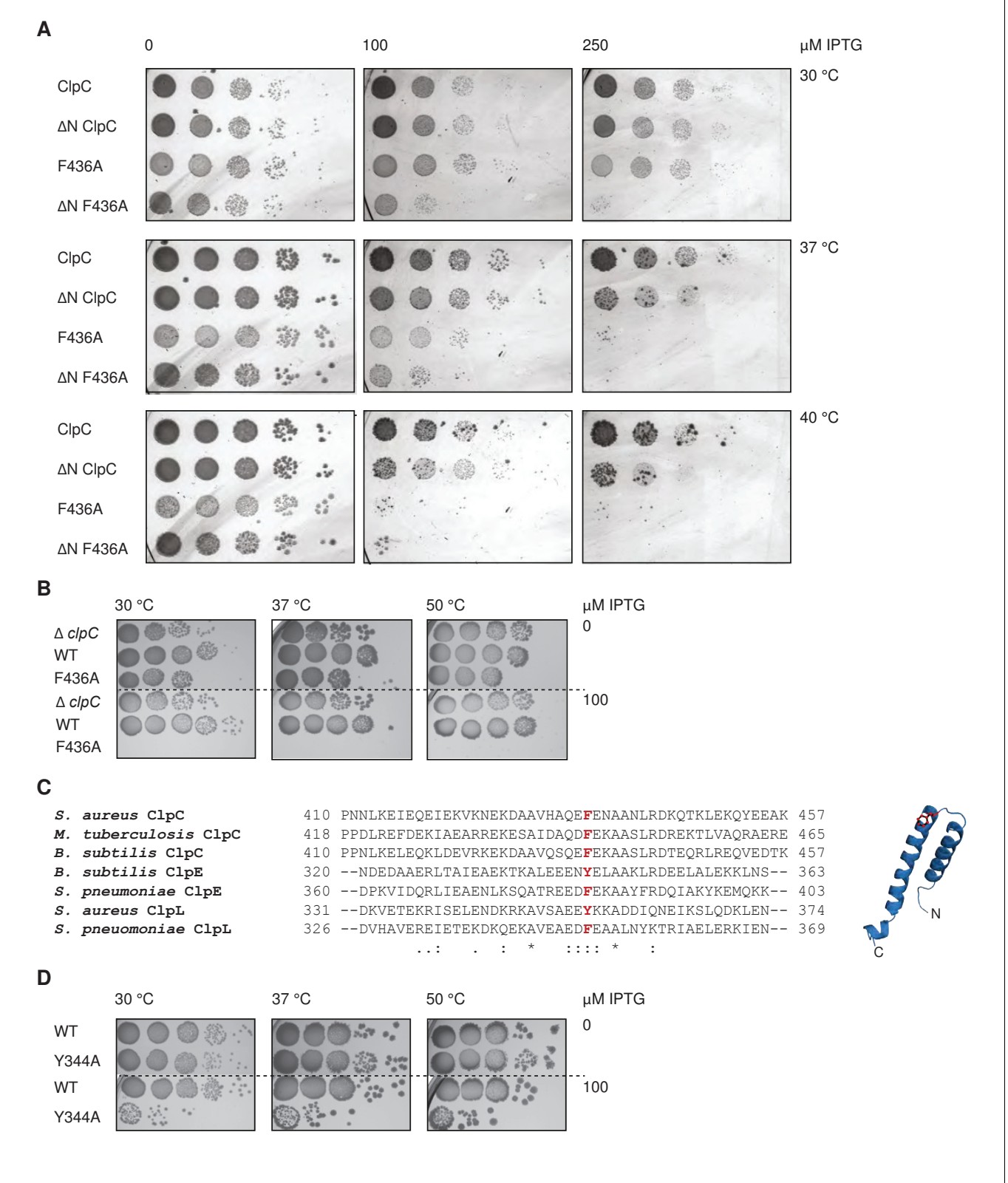

**Figure 6.** Loss of ClpC activity control is toxic in vivo. (**A**) *E. coli* cells constitutively expressing *S. aureus clpP* and harboring the indicated plasmid-encoded *clpC* alleles under control of an IPTG-regulatable promoter were grown overnight at 30°C and adjusted to OD$_{600}$ of 1. Serial dilutions (10$^{-2}$ – 10$^{-6}$) were spotted on LB plates containing the indicated IPTG concentrations and incubated at 30°C, 37°C or 40°C for 24 hr. (**B**) *B. subtilis* ΔclpC control cells and ΔclpC mutant cells harboring a *clpC* wild type (WT) or MD mutant (F436A) copy integrated at the *amyE*-locus under control of an

*Figure 6 continued on next page*

*Figure 6 continued*

IPTG-regulatable promoter were grown at 30℃ to OD600 = 1. Serial dilutions ($10^{-2} - 10^{-6}$) were spotted on LB plates without or with 100 µM IPTG and incubated at 30℃, 37℃ or 50℃ for 24 hr. (C) Sequence alignment of MDs from ClpC, ClpE and ClpL proteins. A highly conserved aromatic residue located at the tip of the coiled-coil structure is highlighted. (D) *B. subtilis* cells harboring an extra *clpE* wild type (WT) or MD mutant (Y344A) copy integrated at the *amyE*-locus under control of an IPTG-regulatable promoter were grown at 30℃ to OD600 = 1. Serial dilutions ($10^{-2} - 10^{-6}$) were spotted on LB plates without or with 100 µM IPTG and incubated at 30℃, 37℃ or 50℃ for 24 hr.
DOI: https://doi.org/10.7554/eLife.30120.019

The following figure supplement is available for figure 6:

**Figure supplement 1.** Expression of toxic ClpC MD mutants.
DOI: https://doi.org/10.7554/eLife.30120.020

Our findings extend the role of coiled-coil MDs as regulatory devices controlling AAA+ protein activity. MDs of the ClpB/Hsp104 disaggregases function as molecular toggles, which are crucial for AAA+ protein repression in the ground state and activation by an Hsp70 partner chaperone (*Oguchi et al., 2012*; *Heuck et al., 2016*; *Rosenzweig et al., 2013*; *Lee et al., 2013*; *Seyffer et al., 2012*) (*Figure 7*). Repression by ClpB/Hsp104 MDs relies on formation of a repressing belt around a canonical AAA+ ring by interacting with AAA-1 domains and neighboring MDs. Intermolecular head-to-tail contacts between long MDs (~120 residues forming two wings) are crucial to keep MDs in a horizontal, repressing conformation (*Carroni et al., 2014*) (*Figure 7*). Hsp70 binding to the tip of one wing breaks MD interactions and leads to ClpB activation.

ClpC MDs cannot function in the same manner due to their reduced size (~50 residues forming a single wing), which is too short to span the distance between neighboring subunits in a hexameric assembly. Instead, ClpC MDs form intermolecular head-to-head contacts allowing docking of two layers of ClpC molecules arranged in a helical conformation (*Figure 7*). The formation of this structure also involves ATP binding to AAA domains. We define this large ClpC assembly as inactive resting state, as it strongly restricts binding of substrates, ClpP and adaptor proteins and does not allow for efficient ATP hydrolysis.

Thereby ClpC MDs function as molecular switches, similar to ClpB/Hsp104 MDs, ensuring repression in the ground state and allowing for activation in presence of substrate recruiting adaptors (*Figure 7*). This dual activity is best illustrated for MD residue F436 located at the tip of the coiled-coil structure. F436 is essential for both, intermolecular MD interaction and MecA binding (*Figure 7*). The N-terminal domain, though not being essential for resting state formation, appears to play an additional role in this switch of conformations, by going from a more hidden position in between MDs in the resting state to a more exposed one, available to MecA or other adaptors.

The interaction surface of MDs is limited (50 Å$^2$) which has implications on the stability of the resting state complex. We speculate that multiple MD contacts provide sufficient stickiness to stabilize a resting state core, while providing dynamics to peripheral subunits, in agreement with our structural and biochemical analysis. As MecA-binding sites in N- and M-domains are not accessible in the resting state, we suggest that ClpC subunit dissociation is prerequisite for MecA binding. Additionally, MecA might bind to peripheral subunits of the ClpC storage state causing their displacement. We, however, do not exclude a scenario in which dissociation of the complete ClpC double-spiral takes place, allowing for more direct transition into hexamers upon MecA-binding to a single ClpC spiral.

Our newly derived model of *S. aureus* ClpC activity control differs from a former one, showing that *B. subtilis* ClpC is monomeric and requires MecA for hexamer formation (*Kirstein et al., 2006*). Sequestering inactive ClpC subunits in a resting state structure might provide better protection against non-specific interactions with other cellular components or degradation by proteases as compared to freely accessible monomers. We realize that former ClpC analysis was performed in presence of high salt concentrations (300 mM NaCl), which likely interfere with MD head-to-head contacts involving charged residues (R443, D444). A regulatory model involving only monomer-hexamer transitions cannot explain activation and severe cellular toxicity of ClpC MD mutants shown here in *E. coli* and *B. subtilis* cells. However, we suggest that the oligomerization dependence of ClpC on MecA at high salt concentrations might represent a fail-safe system ensuring adaptor-dependent ClpC activation under conditions that do not allow for resting state formation (e.g. salt stress).

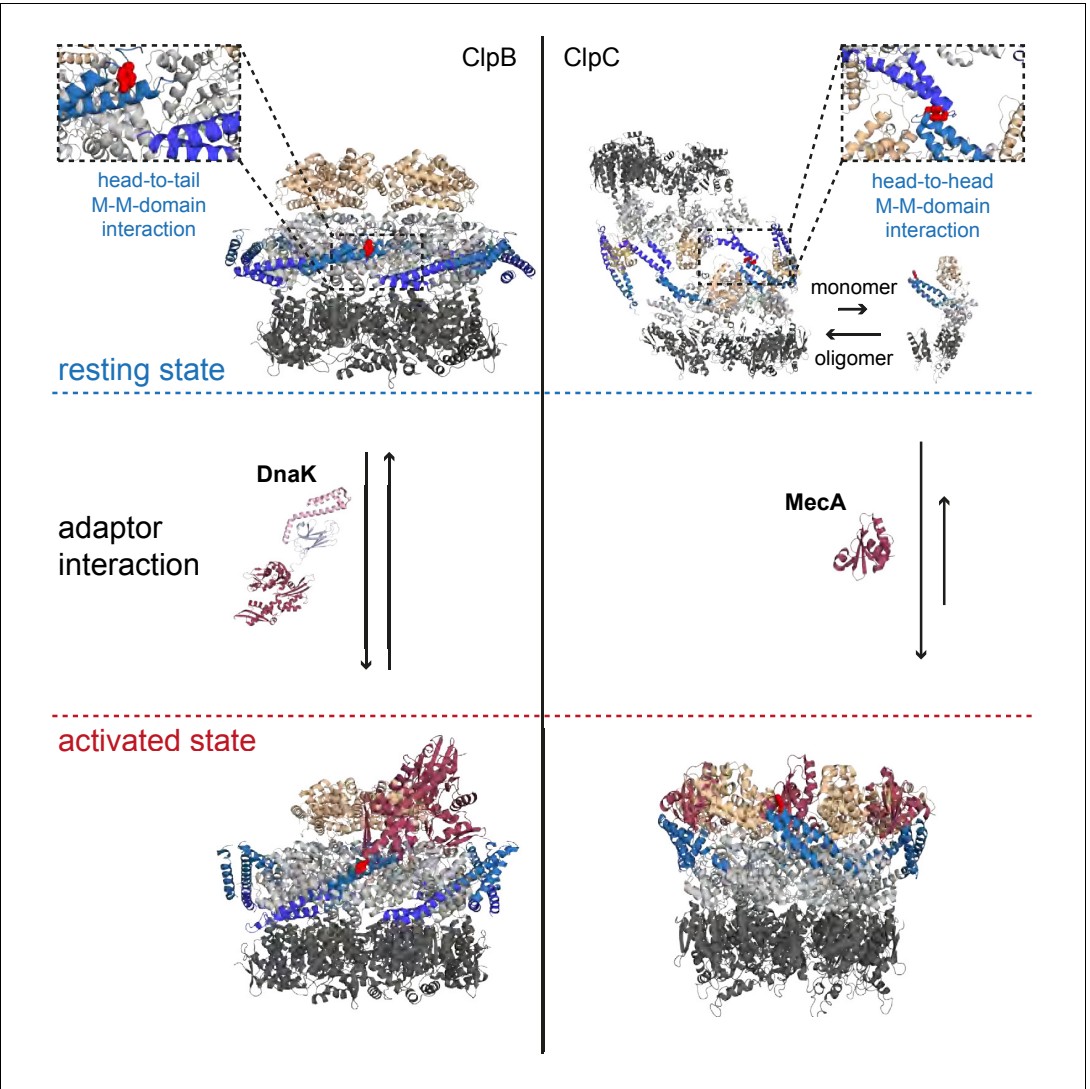

**Figure 7.** Regulatory coiled-coil MDs repress AAA+ protein activities by different mechanisms. ClpB is kept in a low activity resting state by long MDs forming a repressive belt around the hexameric AAA ring. MDs are kept in place by head-to-tail interactions between adjacent coiled-coils. In contrast, the ClpC resting state is formed by head-to-head MD contacts, allowing for assembly of two open ClpC spirals. Adaptor proteins of ClpB (DnaK) and ClpC (MecA) break MD contacts by binding to MD sites crucial for MD interactions. This results in AAA+ protein activation by releasing MD repression on ATPase activity (ClpB) or allowing for formation of active hexamers (ClpC).

DOI: https://doi.org/10.7554/eLife.30120.021

Once formed the hexameric ClpC$_6$/MecA$_6$ complex is stable and does not dissociate spontaneously, raising the question how ClpC activation is turned off. Adaptor proteins are targeting themselves for degradation by ClpC/ClpP if substrates are no longer available (*Mei et al., 2009*; *Turgay et al., 1998*; *Kirstein et al., 2007*). This mechanism couples substrate availability with ClpC activation and ensures fast ClpC inactivation in absence of substrate by causing dissociation of ClpC hexamers into monomers that are subsequently sequestered in the resting state. Notably, other chaperone machineries including the Hsp70 member BIP (*Preissler et al., 2015*) and the AAA+ protein Rca (*Mueller-Cajar et al., 2011*) also form large, inactive resting states that are converted into active species depending on substrate availability. Sequestration of chaperones therefore seems a more widespread activity to tune their activities according to the physiological need.

Constitutively activated ClpC MD mutants exert strong toxicity in *E. coli* and *B. subtilis* cells, demonstrating an essential physiological need to tightly control ClpC function. In contrast, no ClpC toxicity is observed in *B. subtilis* cells expressing ClpC wild type, which unlike ClpC MD mutants can interact with the complete set of its adapter proteins. This suggests that adapters confer a higher degree of substrate specificity to ClpC as compared to activated ClpC MD mutants. We assume that mistargeted protein degradation by deregulated ClpC MD mutants of e.g. newly synthesized proteins or secretory proteins, which did not yet reach their native structures or cellular compartment, leads to cell death.

Homologous MDs are present in the AAA+ protein family members ClpE and ClpL and key residues driving formation of the ClpC resting state are evolutionary conserved. We show that mutating a key regulatory MD residue of ClpE also causes cellular toxicity, strongly suggesting that the repressing mode of MDs established here for ClpC is also operational in ClpE and ClpL and thus is a more general mechanism for controlling bacterial AAA+ chaperone systems.

Severe toxicity of ClpC and ClpE MD mutants qualifies these AAA+ chaperones as targets for antimicrobials. In fact, the deregulation of bacterial proteases represents a novel antibacterial strategy (*Malik and Brötz-Oesterhelt, 2017*; *Culp and Wright, 2017*). Notably, the *M. tuberculosis* ClpC N-terminal domain was recently identified as target of cyclic peptides with antibacterial activities (*Schmitt et al., 2011*; *Gao et al., 2015*; *Gavrish et al., 2014*). Although the mode of these drugs and their effects on ClpC function are not understood it is likely that they interfere with ClpC activity control. Our findings presented here open a new route for toxic ClpC deregulation by identifying MDs as crucial regulatory elements and thus drug targets and offering a promising approach to attack multi-drug resistant bacteria.

## Materials and methods

### Strains, plasmids and proteins

*E. coli* strains used were derivatives of MC4100, XL1-blue or DH5α. ClpC, MecA, ClpP were amplified by PCR, inserted into pDS56 and verified by sequencing. Mutant derivatives of ClpC were generated by PCR mutagenesis and standard cloning techniques in pDS56 and were verified by sequencing. Transformation into *B. subtilis* 168 was performed by standard methods (*Anagnostopoulos and Spizizen, 1961*). AmyE insertion in *B. subtilis* was checked by plating on agar containing 0.4% starch (w/v) additionally to appropriate antibiotics, screening for successful loss of α-amylase by staining starch with Lugol's iodine.

ClpC and variants, MecA and ClpP were purified after overproduction from *E. coliΔclpB::kan* cells. GFP-SsrA was purified after overproduction from *E. coli ΔclpX ΔclpP* cells. All proteins were purified using Ni-IDA (Macherey-Nagel) and size exclusion chromatography (Superdex S200, GE Healthcare) following standard protocols. Pyruvate kinase of rabbit muscle, casein and FITC-casein were purchased from Sigma. Protein concentrations were determined with the Bio-Rad Bradford assay.

### Biochemical assays

#### Size exclusion chromatography and multi-angle light scattering

Complex formation of ClpC (10 µM) was monitored by size exclusion chromatography (SEC, Superose 6 10/300 GL, GE Healthcare). MecA (20 µM), ClpP (20 µM), casein (16.66 µM) and FITC-casein (2.5 µM) were added as indicated. Experiments were run at 25°C in buffer A (50 mM Tris pH 7.5, 25 mM KCl, 20 mM MgCl$_2$) supplemented with 2 mM ATP. Samples were prepared freshly and incubated for 5 min with 2 mM ATP prior to injection. Fractions were collected in 96-well plates, aliquots taken and subjected to SDS-PAGE. Gels were stained using SYPRO Ruby Protein Gel Stain (Thermo-Fisher) following manufacturer instructions. Band intensities were quantified using ImageJ. To monitor the binding of ClpC to FITC-Casein, the collected fractions were analyzed for FITC fluorescence using FLUOstar Omega (BMG Labtech) with standard FITC filter sets. Chromatography was performed in three independent experiments each and representative results are provided.

Complex formation analyzed by SEC was additionally followed by online multi-angle light scattering (MALS) using an Agilent 1260 Infinity II HPLC system connected in series with a 3-angle multi-angle light scattering detector (miniDAWN TREOS II, Wyatt Technology, collection rate of 2 data points per second) and an additional online differential refractive index detector (Optilab T-rEX,

Wyatt Technology) for concentration determination. Data analysis was performed using ASTRA 7.1 (Wyatt Technology). Samples were additionally filtered through a 0.2 μm low-protein binding syringe filter (Millex-GV, Merck Millipore Ltd.) before application to the SEC-column.

## ATPase activity

The ATPase rate of ClpC and mutants was determined using a coupled-colorimetric assay as described before (*Oguchi et al., 2012*). The assay was carried out at 2 mM ATP in buffer A including 2 mM DTT at 30°C using a FLUOstar Omega plate reader. The final protein concentrations were as follows: ClpC (1 μM), MecA (1.5 μM), casein (10 μM). In presence of casein MecA concentrations were reduced to 0.2 μM. The raw data was analyzed using the following equation:

$$\text{ATPase rate} = \frac{1}{\varepsilon(\text{NADH}) \cdot c(\text{ATPase}) \cdot d} \cdot \frac{d(A_{340\,\text{nm}})}{dt}$$

$\varepsilon(\text{NADH})$: Extinction coefficient at 340 nm for NADH ($M^{-1}\,cm^{-1}$)
$c(\text{ATPase})$: Concentration of ATPase (M)
$d$: path length (cm)
$d(A_{340\,\text{nm}})dt$: derivative of the linear graph (slope)

ATPase rates were calculated from the linear decrease of $A_{340}$ in at least three independent experiments and standard deviations were calculated.

## Degradation assays

FITC-casein degradation was analyzed using a CLARIOstar plate reader, in black 384 well plates (Corning, NBS coated, flat bottom), in buffer A with 2 mM DTT. The final protein concentrations were as follows 0.3 μM FITC-casein, 1 μM ClpC, 1.5 μM MecA, 2 μM ClpP. The assay was carried out in the presence of an ATP regenerating system (0.02 mg/mL PK, 3 mM PEP pH 7.5) and 2 mM ATP. The increase of FITC-casein fluorescence upon its degradation was monitored by using 483 and 520/530 nm as excitation and emission wavelengths, respectively.. For data processing the background in the absence of ClpC was subtracted and the initial fluorescence intensities were set to 1. FITC-casein degradation rates were determined by the initial slopes of the fluorescence signal increase in at least three independent experiments and standard deviations were calculated. Alternatively degradation of FITC-casein (5 μM) was monitored by SDS-PAGE followed by Coomassie staining.

Degradation of GFP-SsrA (0.2 μM) was performed in buffer A with 2 mM DTT using the following protein concentrations: 1 μM ClpC, 1.5 μM MecA, 2 μM ClpP. Reactions were started by addition of an ATP regenerating system (0.02 mg/mL PK, 3 mM PEP pH 7.5) and 2 mM ATP. GFP fluorescence was monitored with a LS55 spectrofluorimeter (Perkin Elmer) or a CLARIOstar plate reader (using black 384 well plates, corning, NBS coated, flat bottom) using 400 and 510 nm as excitation and emission wavelengths. Degradation rates were determined by the initial slopes of fluorescence signal decrease in at least three independent experiments and standard deviations were calculated.

For FITC-casein and GFP-SsrA degradation under saturating conditions MecA and ClpP concentrations were increased to 2 and 4 μM, respectively.

## Crosslinking

Glutaraldehyde crosslinking was performed by incubating 1 μM ClpC or ClpB buffer B (50 mM HEPES, 25 mM KCL, 10 mM $MgCl_2$, 2 mM DTT, pH 7.5) in absence or presence of 2 mM ATP/ATPγS and 3 μM MecA at 25°C for 15 min. Crosslinking was started by adding Glutaraldehyde (Sigma) to a final concentration of 0.1%. Aliquots were taken at indicated time points and crosslinking was quenched by adding Tris (pH 7.5) to a final concentration of 50 mM. Samples were subjected to SDS-PAGE and gels stained with SYPRO Ruby Protein Gel Stain (ThermoFisher).

Disulfide crosslinking was performed by incubating 3 μM ClpC in buffer A (in the absence or presence of 2 mM nucleotide). MecA (4.5 μM) was added as indicated MecA and 2 mM β-Mercaptoethanol at 25°C for 5 min. Crosslinking was started by addition of copper-phenanthroline to a final concentration of 100 μM. Aliquots were taken and crosslinking was stopped by adding SDS sample buffer without β-mercaptoethanol but containing 4 mM iodacetamide. Samples were boiled and analyzed by SDS-PAGE followed by Coomassie staining.

Crosslinking was performed in two or more independent experiments each and representative results are provided.

## Anisotropy measurements

FITC-casein (100 nM) was incubated with varying concentrations ClpC in buffer A with 2 mM DTT for 1 hr in absence or presence of 2 mM ATPγS. Changes in fluorescence polarization were determined using a CLARIOstar plate reader (BMG Labtech) at 482 and 530 nm excitation and emission wavelengths (Target mP 35).

## Western blotting

SDS-PAGEs were transferred to nitrocellulose or PVDF membranes by semi-dry blotting or wet blot transfer. Membranes were subsequently blocked with either 3% BSA (w/v) or 5% (w/v) skim milk powder in TBS-T. Custom-made antibodies were used at the following dilutions: anti-ClpC (*B. subtilis*) 1:100.000, anti-ClpE 1:30.000, anti-ClpC (*S. aureus*) 1:50.000 and anti-MecA 1:30.000. anti-rabbit alkaline phosphatase conjugate (Vector Laboratories) was used as secondary antibody (1:10.000). Blots were developed using NBT/BCIP or ECF Substrate (GE Healthcare) as reagent and imaged via Image-Reader LAS-4000 (Fujifilm). Western blotting was performed in two or more independent experiments each and representative results are provided.

## ClpC structure determination by cryo-electron microscopy

### Cryo-electron microscopy

*S. aureus* ClpC WT (6 μM) was incubated for 15 min at room temperature in 25 mM Tris-HCl (pH 7.5), 25 mM KCl, 10 mM $MgCl_2$, 1 mM DTT and 2 mM ATPγS. For ClpC-MecA complex formation, *S. aureus* MecA was incubated with ClpC in a 3:1 molar ratio. Samples were vitrified with liquid ethane on Quantifoil R2/2 grids using a Vitrobot Mark IV (FEI) at 100% humidity, 24°C temperature and blotting time of 3 s.

Images of ClpC were collected using the EPU software on a Titan Krios TEM (FEI) operating at 300kV, using a Falcon two direct electron detector (FEI). Images of ClpC in complex with MecA were collected using the EPU software on a Titan Krios TEM (FEI) operating at 300kV equipped with a Gatan K2 Summit direct electron detector and bioquantum energy filter with 20 eV slit. The defocus range was set between −1 and −3 μm with a total dose of 30 electrons/Å (*Doyle et al., 2013*) in 17 frames for ClpC and 50 electrons/Å (*Doyle et al., 2013*) in 40 frames for ClpC-MecA. Pixel size was 1.34 Å/pixel for ClpC and 1.37 Å/pixel for ClpC-MecA. The dose rate on the K2 camera was 4.125 e/pixel/sec and the exposure time 23 s.

### Image processing

Movie frames alignment with dose weighting (*Zheng et al., 2017*) and CTF estimation (*Rohou and Grigorieff, 2015*) was performed on-the-fly using a Scipion suite (*de la Rosa-Trevín et al., 2016*). ClpC particles were picked with Gautomatch and a dataset of ~90.000 particles from 1100 micrographs was generated. ClpC-MecA complex particles were picked using Gaussian picking in RELION (*Scheres, 2012*) and ~500.000 particles from 2100 micrographs were obtained. The initial datasets were subjected to reference-free 2D classification in order to clean the datasets.

Initially, for 3D processing of ClpC (Supplementary *Figure 4*), the crystal structure of ClpC-MecA (*Wang et al., 2011*) low-pass filtered at 60 Å was used, but the dataset failed to refine. Attempts to use *ab initio* models generated with Eman2 (*Tang et al., 2007*) also did not result in high-resolution 3D refinement. As the ClpC assembly appeared much larger in size than the ClpC-MecA hexamer, we generated a cylindrical starting model by filtering to 70 Å two copies of ClpC-MecA stuck back to back with the AAA2 rings in contact. With this starting model ClpC refined to 8.4 Å resolution as estimated with the 0.143 FSC criterion, with visible separated helices. The same result was confirmed by generating an *ab-initio* starting model using the SGD method implemented in cryoSPARC (*Punjani et al., 2017*) and refining the structure within the same program suite. Reconstructions with and without C2 symmetry applied were performed (Supplementary *Figure 4*, g-h) to a similar resolution and the C2 map was used for display as it allows a better visualization of N-domains.

For 3D processing of ClpC-MecA (Supplementary *Figure 5*) the crystal structure of the *B. subtilis* ClpC-MecA complex filtered at 60 Å was used (pdb code: 3PXI). A large dataset was initially used,

but particles were preferentially oriented so a reduced dataset (~26.000 particles) with balanced angular distribution was used to reduce anisotropy. Both asymmetric and six fold symmetric maps were built. Even though the nominal resolution of the symmetrized map was better, the reconstruction appeared over filtered, thus indicating that artifacts were caused by forced symmetrisation. The final map was reconstructed at 11 Å resolution. Local resolution was evaluated using the local resolution tool of RELION.

## Model building and fitting

The *S. aureus* atomic model of ClpC and MecA monomers were generated using Phyre2 (*Kelley and Sternberg, 2009*). They were modelled based on the *B. subtilis* atomic structures (PDBcode: 3pxi). Each protomer was rigidly fitted manually using USCFChimera (*Pettersen et al., 2004*) and refined using iModFit (*Lopéz-Blanco and Chacón, 2013*). For the ClpC resting state structure, real-space refinement in Phenix and Rosetta was performed trying to optimise as much as possible the model geometry and giving less weight to the map. Flexible, non-fittable loops where cut away from the final deposited pdb (PDBcode: 6EM9). Fittings were performed both on the C2 symmetrised and asymmetric maps and both maps and C-α coordinates have been deposited (EMD-3895, EMDB-3894). For the ClpC-MecA complex the *S. aureus* Phyre model was fitted into the map using Chimera and refined with iMODfit. The ClpC-MecA map and the fitted pdb (only C-alpha) have been deposited (PDBcode: 6EMW).

Analysis of AAA domain subunit interfaces was performed using PISA (http://www.ebi.ac.uk/pdbe/pisa/picite.html).

## Spot tests

*E. coli* cells harboring plasmid-encoded *clpC* alleles were grown in the absence of IPTG overnight at 30°C. Serial dilutions were prepared, spotted on LB-plates containing different IPTG concentrations and incubated for 24 hr at indicated temperatures. *B. subtilis* strains were inoculated with a fresh overnight culture to an $OD_{600}$ of 0.05 and grown to mid-exponential growth phase. Optical densities of all strains were adjusted to $OD_{600}$ of 1, serial dilutions were performed and 10 μl ($10^{-2}$ - $10^{-6}$) were dropped on agar plates (without or with 100 μM IPTG) and incubated overnight at indicated temperatures. Spot tests were performed in two or more independent experiments each and representative results are provided.

## Accession numbers

The EM maps have been deposited in the 3D-EM database (www.emdatabank.org) with accession codes EMD-3895 (ClpC resting state C1), EMDB-3894 (ClpC resting state C2) and EMD-3897 (ClpC-MecA). Half maps and masks have also been deposited. PDB models (including only C-α) based on the EM maps have also been deposited with codes 6EM9 (ClpC resting state C1), 6EM8 (ClpC resting state C2) and 6EMW (ClpC-MecA).

## Acknowledgements

KBF was supported by the Hartmut Hoffmann-Berling International Graduate School of Molecular and Cellular Biology (HBIGS). This work was funded by grants of the Deutsche Forschungsgemeinschaft (BB617/17-2 and MO 970/4–2) to BB and AM and a fellowship of the Hannover School for Biomolecular Drug Research to IH and the DFG grants Tu106/8-1 and Tu106/6-2 to KT. The Cryo-EM facility at the Science for Life Laboratory Stockholm University (MC) is supported by grants from the Knut and Alice Wallenberg Foundation and the Family Erling Persson Foundation. We thank Stefan Fleischmann for IT support, Christos Savva for microscopy support, Björn Forsberg and Shintaro Aibara for image-processing discussions, Helen Saibil for support in the early stage of the project and Armgard Janczikowski for technical assistance.

## Additional information

### Funding

| Funder | Grant reference number | Author |
|---|---|---|
| Deutsche Forschungsge-meinschaft | BB617/17-2 | Bernd Bukau<br>Axel Mogk |
| Hartmut Hoffmann-Berling International Graduate School of Molecular and Cellular Biology | | Kamila B Franke |
| Hannover School for Biomolecular Drug Research | | Ingo Hantke |
| Knut och Alice Wallenbergs Stiftelse | | Marta Carroni |
| Familjen Erling-Perssons Stiftelse | | Marta Carroni |
| Deutsche Forschungsge-meinschaft | MO970/4-2 | Bernd Bukau<br>Axel Mogk |
| Deutsche Forschungsge-meinschaft | Tu106/8-1 | Kürşad Turgay |
| Deutsche Forschungsge-meinschaft | Tu106/6-2 | Kürşad Turgay |

The funders had no role in study design, data collection and interpretation, or the decision to submit the work for publication.

### Author contributions

Marta Carroni, Conceptualization, Formal analysis, Supervision, Funding acquisition, Investigation, Writing—original draft, Project administration; Kamila B Franke, Conceptualization, Data curation, Formal analysis, Investigation, Writing—original draft; Michael Maurer, Formal analysis, Investigation, Writing—review and editing; Jasmin Jäger, Ingo Hantke, Felix Gloge, Daniela Linder, Sebastian Gremer, Kürşad Turgay, Formal analysis, Investigation; Bernd Bukau, Formal analysis, Writing—review and editing; Axel Mogk, Conceptualization, Formal analysis, Writing—original draft

### Author ORCIDs

Sebastian Gremer (ID) http://orcid.org/0000-0003-3421-8449
Kürşad Turgay (ID) http://orcid.org/0000-0002-8959-492X
Axel Mogk (ID) http://orcid.org/0000-0003-3674-5410

### Decision letter and Author response
Decision letter https://doi.org/10.7554/eLife.30120.035
Author response https://doi.org/10.7554/eLife.30120.036

## Additional files

### Supplementary files
• Transparent reporting form
DOI: https://doi.org/10.7554/eLife.30120.022

### Major datasets
The following datasets were generated:

| Author(s) | Year | Dataset title | Dataset URL | Database, license, and accessibility information |
|---|---|---|---|---|
| Marta Carroni, Axel Mogk, Kamila B Franke, Bernd Bu-kau | 2017 | Staphylococcus aureus ClpC resting state, C2 symmetrised | http://www.rcsb.org/pdb/search/structid-Search.do?structureId=6EM8 | Publicly available at Protein Data Bank (accession no: 6EM8) |
| Marta Carroni, Axel Mogk, Kamila B Franke, Bernd Bu-kau | 2017 | Staphylococcus aureus ClpC resting state, asymmetric map | http://www.rcsb.org/pdb/search/structid-Search.do?structureId=6EM9 | Publicly available at Protein Data Bank (accession no: 6EM9) |
| Marta Carroni, Axel Mogk, Kamila B Franke, Bernd Bu-kau | 2017 | Structure of Staphylococcus aureus ClpC in complex with MecA | http://www.rcsb.org/pdb/search/structid-Search.do?structureId=6EMW | Publicly available at Protein Data Bank (accession no: 6EMW) |

The following previously published datasets were used:

| Author(s) | Year | Dataset title | Dataset URL | Database, license, and accessibility information |
|---|---|---|---|---|
| Heuck A, Schitter-Sollner S, Clausen T | 2016 | Crystal structure of Hsp104 | http://www.rcsb.org/pdb/search/structid-Search.do?structureId=4D4W | Publicly available at Protein Data Bank (accession no: 5D4W) |
| Wang F, Mei ZQ, Wang JW, Shi YG | 2011 | Structure of MecA108:ClpC | http://www.rcsb.org/pdb/search/structid-Search.do?structureId=3PXI | Publicly available at Protein Data Bank (accession no: 3PXI) |
| Gates SN, Yokom AL, Lin J-B, Jackrel ME, Rizo AN, Kend-sersky NM, Buell CE, Sweeny EA, Chuang E, Torrente MP, Mack KL, Su M, Shorter J, South-worth DR | 2017 | S. cerevisiae Hsp104-ADP complex | http://www.rcsb.org/pdb/search/structid-Search.do?structureId=5VY8 | Publicly available at Protein Data Bank (accession no: 5VY8) |

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
