## [Decision Letter]

Thank you for submitting your article "Regulatory coiled-coil domains promote head-to-head assemblies of AAA+ chaperones essential for tunable activity control" for consideration by *eLife*. Your article has been favorably evaluated by Ivak Dikic (Senior Editor) and three reviewers, one of whom, Andreas Martin (Reviewer #1), is a member of our Board of Reviewing Editors. The following individuals involved in review of your submission have agreed to reveal their identity: Eilika Weber-Ban (Reviewer #2); Gabriel C Lander (Reviewer #3).

The reviewers have discussed the reviews with one another and the Reviewing Editor has drafted this decision to help you prepare a revised submission.

Summary:

In this manuscript, Carroni and Franke et al. identify and characterize the resting state of *Staphylococcus areus* ClpC, in which the coiled coil middle domains (MDs) mediate a head-to-head interaction of two pentameric ClpC lockwashers. Comparison of cryo-EM reconstructions for ClpC in the absence and presence of the adaptor MecA gives intriguing structural insights into potential mechanisms for ClpC activation, where the disruption of the MD coiled-coil interactions by MecA splits the ClpC lockwashers apart and induces their reconfiguration into hexamers with robust ATPase and protein unfoldase activity.

The combination of these structural data with mutagenesis, biochemical analyses, and in vivo studies allows the proposal of a compelling model about the role of MDs in regulating the ClpC oligomeric state, MecA adaptor interactions, and the proteolytic activity of the ClpCP protease. So far, only structures of the active, MecA-bound ClpC hexamer had been available, whereas the mechanisms underlying ClpC's inactivation in the absence of MecA remained elusive. The present study thus significantly advances our understanding of ClpC regulation and unravels a new concept of AAA motor regulation by coiled coil middle domains.

Essential revisions:

1) A major concern regarding the biochemical experiments is that none of the presented degradation assays were performed under true steady state or multiple turnover conditions. The authors used either 0.3 μm FITC casein or 0.2 μm GFP-SsrA substrate with 1 μm ClpC and 2 μm ClpP. Neither in the figure legends nor the Materials and methods it is specified whether the ClpC and ClpP concentrations refer to monomers or hexamers/tetradecamers, but assuming monomers, those concentrations would be equivalent to 166 nM ClpC6 and 142 nM ClpP14. There is not even enough ClpP to saturate all ClpC, and the substrate concentration is barely above the enzyme concentration. Consequently, degradation kinetics (e.g. in Figure 1) strongly resemble exponential behavior, as expected for single-turnover reactions. Measuring the initial slopes of these curves, as done by the authors, may give a qualitative impression of enzyme activity, but is certainly not suited to determine quantitative degradation rates.

In addition, the substrate affinities of WT ClpC and its various MD and N-domain mutants remain completely elusive, and it is unclear whether the used concentrations reflect saturating conditions. This is particularly important when comparing ClpC mutations that differentially affect the ATPase activity or substrate affinity. For instance, it is proposed that the ClpC N-domain interferes with GFP-SsrA binding and therefore degradation. However, N-terminal deletion also leads to an almost 20-fold stimulation of WT ATPase activity and a > 2-fold stimulation for the F436A and R443A mutants. It therefore remains unclear whether the N-terminal domain indeed inhibits GFP-SsrA binding or makes GFP unfolding less efficient due to lower ATPase activity.

In general, performing these measurements at just a single substrate concentration is certainly not sufficient to quantitatively compare the degradation activities of ClpC variants with mutations that differentially affect Km and Vmax.

To draw reliable conclusions about substrate binding and ClpC motor activity, the authors should perform Michaelis-Menten analyses for at least a couple of key mutants presented, and otherwise should perform multiple-turnover measurements with saturating concentrations of substrate to derive more reliable degradation rates. The reviewers agreed that this should be easy to accomplish with the FITC-casein and especially the GFP-SsrA model substrates that can be produced in high amounts.

2) Similarly, the concentrations used for measuring the stimulatory effects of casein substrate on the ATPase activity of WT ClpC/P are not ideal for quantitative analyses (Figure 2). Even though the authors confirmed a 1:1 complex of ClpC and MecA, they used only 0.2 μm MecA with 1 μm ClpC and 10 μm casein. Under these conditions, 80% of ClpC subunits would not be bound to MecA. In addition, the authors again used only 2 μm ClpP monomers, equivalent to 142 nM ClpP14 (while there is 166 nM ClpC6 present). Having not all ClpC saturated with ClpP and MecA thus leads to strongly convoluted ATPase rates, with contributions from free ClpC, ClpC/P, MecA-ClpC/P, and substrate bound MecA-ClpC/P, which doesn't allow accurate quantitative conclusions about the stimulatory effects of substrate. The authors should therefore repeat those experiments under saturating conditions, both for substrate and MecA.

3) The fact that the lockwasher is pentameric and not hexameric is intriguing – there does appear to be disordered density that might correspond to a sixth subunit in Figure 3—figure supplement 1. No mention is made of this density, and the reviewers wondered if the authors tried low-pass filtering the density or viewing the density at low iso-surface thresholds to assess the possibility that this is a transiently or flexibly associated sixth subunit. The SLS data suggest that the complex is decameric, but this region should nonetheless be investigated through further cryoEM analysis, such as by focused classification using a 3D mask in this area. There may be a subset of particles that more clearly resolved density. Or does the conformation sterically prevent the association of an additional subunit? This should be explored in more detail.

4) Based on their finding that the F436A-DWB mutant stayed monomeric in the absence of nucleotide, the authors propose that the ClpC decameric structure relies entirely on MD contacts. This is surprising, and given the presented structural data (Figure 4) it is hard to imagine how the MDs could provide sufficient lateral interactions to hold neighboring ClpC subunits together. Is there indeed no contribution of the AAA1 and AAA2 domains, which seem to have much more extensive interactions than the MD and neighboring N-domains?

Do the authors have information about whether or not deltaN ClpC can form an inactive higher order resting state like wt ClpC? Based on the structures, this seems unlikely, as the N-domains feature prominently in achieving the resting state.

In general, the authors should attempt to clarify the role of a potential interplay between N-domains and M-domains, and how much the effects observed for MD deletion or mutation might in fact stem from a MecA-mediated association of MD with the N-domain. Could the viability defects observed in vivo not also be explained by the fact that without MD, MecA cannot keep the N-domains away to the side?

5) Based on the observed effects of ClpC MD mutants on cellular viability, the authors propose that the M-domains are essential to control ClpC/P degradation activity by inactivating ClpC in the absence of MecA. The authors should test and confirm this model by co-expression of WT ClpC and MecA, which is expected to have the same phenotype/toxicity as delta-N ClpC F436A (unless MecA-bound WT ClpC has a much more restricted substrate specificity).

Major points:

1) Flexible fitting was stated as being used to generate the atomic model of the ClpC-MecA structure, but very few details are given regarding its generation, aside from the fact that iModFit was used. Why was flexible fitting not used to generate a model of the ClpC double lockwasher, which is at higher resolution? The fitting shown in Figure 3 has much of the atomic model out of density, and should not be used for detailed structural interpretation. For example, in the second paragraph of the subsection “Head-to-head interactions of M-domains mediate formation of an inactive ClpC resting state”, the authors state that the trans-acting arginine fingers are displaced away from the nucleotide binding pockets (Figure 3—figure supplement 1), but given the limited resolution of the map and poor fitting of the atomic model, this cannot be claimed. The reviewers agreed that it is fine to include side chains in the models depicted in Figure 3 and Figure 4, as these data are supported by biochemistry. However, when depositing atomic models based on the presented intermediate resolution EM, the models should only include the C-alpha's of amino acids.

2) The ClpC-MecA structure is described as "asymmetric", but is the structure organized as a spiral, as has been shown in numerous other AAA ATPases? If not, this is novel and should be described in more detail.

Also, the structures of Hsp104 and VAT in a steep lockwasher-like conformation were recently solved – are there any structural similarities with the resting state of ClpC?

3) It's puzzling that 3D classification was not performed at any point – this is regularly used before 3D refinement of a structure to identify a set of conformationally and compositionally homogeneous particles for processing, and then a classification without further alignment is performed after refinement to identify the subset of particles containing the highest resolution information. If these steps were performed and all 3D classes looked identical, this should be stated. Furthermore, it isn't clear which final density from Figure 3—figure supplement 1 was used for the structural analyses. The final structure from cryoSparc appears to be at higher resolution than the RELION structures – what was the reported resolution of this structure? Was C2 symmetry applied?

4) The authors use glutaraldehyde crosslinking to analyze how MecA and the F436A mutation affect the formation of the decameric resting state of ClpC (Figure 5). Surprisingly, lane 9 shows that most of the double-Walker B mutant still forms the decamer (or even larger structure) in the presence of MecA. Was this trend also observed in the MecA-bound EM sample? In the Discussion, the authors state that "once formed, the ClpC6/MecA6 complex is stable and does not dissociate spontaneously". Why then does MecA-bound ClpC-DWB show any crosslinking larger than hexamers?

5) It is surprising that, based on gel filtration results (Figure 5), the formation of ClpC decamers is ATP-dependent, whereas hexamer formation is not, and the authors should try to discuss this. Furthermore, how does the limited interaction surface of MDs (~ 50 A2) compare to the AAA interfaces? Is this interaction indeed substantial and strong enough to disrupt the interactions in a planar, ATP-bound AAA ring when MecA is absent?

6) MecA has been shown in *B. subtilis* to promote assembly from a lower assembly state (monomeric/dimeric) to the active hexamer. Although the authors convincingly show the existence of an inactive higher order assembly of *S. aureus* ClpC, it remains unclear why such a complex is beneficial. This should be discussed.

The authors suggest that activation upon MecA association occurs via monomeric ClpC. Can they provide evidence for that? Can it be excluded that the double-spiral dissociates into two single spirals that then transition more directly into the hexameric state?

7) It is proposed that the MecA adaptor is degraded and ClpC consequently inactivated when substrates are no longer available. However, the strong cytotoxicity of MD mutants in *E. coli* would suggest that the substrate specificity of ClpC/P is rather broad. The authors also speculate that deregulated ClpC/P may go after newly synthesized proteins, which raises the question whether ClpC/P would indeed ever run out of substrates to then be inactivated. Is the assumption that MecA makes ClpC/P more specific and less promiscuous than MD mutants like F436A? The authors should try to address this in their Discussion.

---

## [Author Response]

Essential revisions:1) […] To draw reliable conclusions about substrate binding and ClpC motor activity, the authors should perform Michaelis-Menten analyses for at least a couple of key mutants presented, and otherwise should perform multiple-turnover measurements with saturating concentrations of substrate to derive more reliable degradation rates. The reviewers agreed that this should be easy to accomplish with the FITC-casein and especially the GFP-SsrA model substrates that can be produced in high amounts.

We addressed this major concern of the reviewers and determined degradation rates in presence of increasing substrate concentrations (FITC-casein, GFP-SsrA). For these experiments, we also increased the concentrations of MecA and ClpP to 2 and 4 μM (monomer), respectively, representing a 2- and 1,7-fold excess over hexameric ClpC (1 μM monomer). These new data (Figure 1—figure supplement 1; Figure 1—figure supplement 3) confirm that the ClpC F436A M-domain mutation causes full activation as the mutant degrades substrates under saturating conditions with similar efficiencies as compared to ClpC wild type in presence of MecA. To generalize this finding, the degradation activities of further ClpC M-domain mutants were analyzed under substrate-saturating conditions (FITC-casein) or in presence of 30-fold excess of substrate (5 μM GFP-SsrA) (new Figure 1—figure supplement 1; Figure 1—figure supplement 3). Together these findings substantiate our former conclusion that ClpC M-domain mutants exhibit high and adaptor-independent activities.

We tried to analyze the impact of N-domains on GFP-SsrA binding but could not monitor interaction of GFP-SsrA with either ClpC wild type (plus MecA), ΔN-ClpC or ΔN-ClpC-F436A, neither by size exclusion chromatography nor fluorescence anisotropy or thermophoresis. Eventually initial unfolding of GFP-SsrA is required for tight binding, which is not possible in our binding assays performed in presence of ATPγS. A function of N-domains in preventing GFP-SsrA binding therefore remains to be explored. We cannot rule out that the two-fold higher ATPase activities of ΔN-ClpC M-domain mutants (as compared to ΔN-ClpC) allow for GFP-SsrA unfolding and now add his possible explanation to the revised manuscript. The finding that the increased ATPase activity of ΔN-ClpC does not allow for GFP-SsrA degradation, however, suggests that the consequences of N- and M-domain deletions/mutations on ClpC are qualitatively different. We show that ΔN-ClpC is still crosslinked to high molecular weight products in presence of ATPγS, whereas ΔN-ClpC-F436A only forms hexameric assemblies (new Figure 5). This confirms the dominant role of M-domains in ClpC resting state formation and activity control.

2) Similarly, the concentrations used for measuring the stimulatory effects of casein substrate on the ATPase activity of WT ClpC/P are not ideal for quantitative analyses (Figure 2). Even though the authors confirmed a 1:1 complex of ClpC and MecA, they used only 0.2 μm MecA with 1 μm ClpC and 10 μm casein. Under these conditions, 80% of ClpC subunits would not be bound to MecA. In addition, the authors again used only 2 μm ClpP monomers, equivalent to 142 nM ClpP14 (while there is 166 nM ClpC6 present). Having not all ClpC saturated with ClpP and MecA thus leads to strongly convoluted ATPase rates, with contributions from free ClpC, ClpC/P, MecA-ClpC/P, and substrate bound MecA-ClpC/P, which doesn't allow accurate quantitative conclusions about the stimulatory effects of substrate. The authors should therefore repeat those experiments under saturating conditions, both for substrate and MecA.

We initially titrated MecA concentrations in ATPase measurements to determine the MecA concentration that is sufficient for maximal stimulation of ClpC ATPase activities (1.5-fold excess of MecA, see Author response image 1). We therefore consider our condition to determine ClpC ATPase stimulation by MecA to be well chosen. These initial assays were additionally performed in absence and presence of ClpP to test for potential impact of the associated peptidase on ClpC ATPase activity, which we found is marginal (see Author response image 1). To avoid degradation of MecA and therefore reduction of ClpC ATPase stimulation during experiments we performed all subsequent ATPase activity measurements in absence of ClpP.

The reviewers are referring to an experiment that aimed at determining the impact of substrate (casein) binding on ClpC ATPase activity (Figure 2). In this particular experiment, we used lower MecA concentrations on purpose as MecA has dual roles by (i) serving as adapter that is delivering substrates to ClpC and (ii) acting as substrate at the same time as evident from its ClpC/ClpP-mediated degradation. To predominantly monitor the impact of MecA-mediated substrate transfer on ClpC ATPase activity, we lowered the MecA concentrations in this setup (0.2 μM) while offering casein excess (10 μM) at the same time. We have clarified this point in the revised manuscript. The reduced MecA concentrations still mediate efficient casein degradation, demonstrating that the adaptor is not limiting at the given concentration.

**Author response image 1. respfig1:** ATPase activity of ClpC (0.5 μM) was determined in presence of increasing MecA concentrations in absence and presence of ClpP (1.5 μM). The ratio of MecA to ClpC is indicated. In this study ClpC ATPase stimulation by MecA was typically performed at 1.5x excess of MecA.

3) The fact that the lockwasher is pentameric and not hexameric is intriguing – there does appear to be disordered density that might correspond to a sixth subunit in Figure 3—figure supplement 1. No mention is made of this density, and the reviewers wondered if the authors tried low-pass filtering the density or viewing the density at low iso-surface thresholds to assess the possibility that this is a transiently or flexibly associated sixth subunit. The SLS data suggest that the complex is decameric, but this region should nonetheless be investigated through further cryoEM analysis, such as by focused classification using a 3D mask in this area. There may be a subset of particles that more clearly resolved density. Or does the conformation sterically prevent the association of an additional subunit? This should be explored in more detail.

We agree with the reviewers, there might be additional subunits in the open part of the double helix, either transiently or flexibly associated. The lockwasher is not strictly pentameric as there is not full density for complete periphery subunits. We provide a new Figure 3—figure supplement 2 with a gallery of the resting state subunits and the visible domains for each of them in the asymmetric reconstruction. In the asymmetric reconstruction, there are 4 full subunits, 4 subunits missing density only for the N-terminus and 2 subunits missing most density for both N-terminus and large AAA1 subdomains. We now discuss the heterogeneity of resting state subunits more precisely in the revised manuscript. In the symmetrized C2 maps the N-termini can be better resolved and there are 6 full subunits + 4 peripheral incomplete ones.

Additionally, we have tried to identify additional subunits within the disordered region by performing focused classification with signal subtraction on the “periphery” of the resting state. It was, however, not possible to separate a population with additional or more defined subunits in the periphery. Focused 3D classification was performed both with and without local alignment. We cannot exclude that a much larger dataset could help to isolate populations were peripheral subunits are more defined.

**Author response image 2. respfig2:** Four classes were generated after 3D focused classification with signal subtraction of the ClpC resting state periphery.

To figure out whether resting state sterically prevents the incorporation of further subunits we generate a model of a closed double spiral. Applying the same shift and rotation that define the helical pitch new subunits were added to generate a closed double spiral. Up to 3 full subunits (with all subdomains) can be added plus an incomplete one (lacking the N-terminus that would otherwise clash). Thus a closed double spiral could theoretically include up to 14 subunits (see Author response image 3). We, however, exclude that this closed conformation stably exists in solution as the ab-initio SGD procedure in cryoSPARC only generated one single model, the open double helix, even when the option of generating 3 starting models was given. We have added a clarification to the manuscript stating that the resting state does not allow for infinite subunit incorporation as observed in e.g. Hsp104 crystal arrangements (subsection “Head-to-head interactions of M-domains mediate formation of an inactive ClpC resting state”, second paragraph).

**Author response image 3. respfig3:** Model of a closed ClpC double-helix generated by applying helical parameters. In magenta are protomers for which there is some missing density in the cryo-EM map (either missing the N-terminus or small AAA subdomains). In dark cyan are protomers completely generated by applying the helical parameters.

4) Based on their finding that the F436A-DWB mutant stayed monomeric in the absence of nucleotide, the authors propose that the ClpC decameric structure relies entirely on MD contacts. This is surprising, and given the presented structural data (Figure 4) it is hard to imagine how the MDs could provide sufficient lateral interactions to hold neighboring ClpC subunits together. Is there indeed no contribution of the AAA1 and AAA2 domains, which seem to have much more extensive interactions than the MD and neighboring N-domains?Do the authors have information about whether or not deltaN ClpC can form an inactive higher order resting state like wt ClpC? Based on the structures, this seems unlikely, as the N-domains feature prominently in achieving the resting state.In general, the authors should attempt to clarify the role of a potential interplay between N-domains and M-domains, and how much the effects observed for MD deletion or mutation might in fact stem from a MecA-mediated association of MD with the N-domain. Could the viability defects observed in vivo not also be explained by the fact that without MD, MecA cannot keep the N-domains away to the side?

We agree with the reviewers and performed additional experiments to clarify the raised issue. We analyzed the impact of N-domains on ClpC resting state formation by glutaraldehyde crosslinking (new Figure 5). In absence of ATP we hardly detect formation of large ΔN-ClpC complexes in contrast to ClpC wild-type pointing to a role of N-domains in stabilizing the resting state. This could be explained by N-domains interacting with neighboring AAA-1 domains in the resting state model. A major difference in crosslinking patterns between ΔN-ClpC and ClpC was, however, no longer apparent in presence of ATPγS as ΔN-ClpC was still crosslinked, to high molecular weight complexes. This indicates, as also pointed out by the reviewers, that contacts between neighboring, ATP-bound AAA-1 and AAA-2 domains play crucial roles for resting state formation which is now discussed in the revised manuscript. Analysis of ΔN-ClpC-F436A (+ ATPγS) revealed that the M-domain mutant does not adopt the resting state but readily forms hexameric assemblies (new Figure 5). This demonstrates a dominant, N-domain independent role of M-domains in controlling ClpC oligomerization and activity. Our new data argue against a crucial interplay of N- and M-domains in ClpC activity control (subsection “Obstructing head-to-head M-domain contacts allows for ClpC hexamer formation”, second paragraph). In agreement, deleting N-domains in ΔN-ClpC does not cause ClpC activation (Figure 1/E).

The reviewers are correct that the interaction surface between M-domains is limited (50 Å2 as pointed out in the original manuscript). We suggest that multiple MD contacts provide sufficient energy and stickiness to promote this particular resting state formation. At the same time, we suggest that the small-sized MD interaction site provides sufficient dynamics to the resting state allowing for constant ClpC subunit association and dissociation. This point has been clarified in the Discussion section (fifth paragraph).

In presence of MecA N- and M-domains do not interact as evident from the ClpC-MecA co-crystal structure. We therefore consider a scenario in which M-domains are required to push N-domains away to the side to prevent toxic ClpC activation as highly unlikely.

5) Based on the observed effects of ClpC MD mutants on cellular viability, the authors propose that the M-domains are essential to control ClpC/P degradation activity by inactivating ClpC in the absence of MecA. The authors should test and confirm this model by co-expression of WT ClpC and MecA, which is expected to have the same phenotype/toxicity as delta-N ClpC F436A (unless MecA-bound WT ClpC has a much more restricted substrate specificity).

We followed the suggestion of the reviewers and simultaneously expressed *S. aureus* ClpP, MecA and ClpC in *E. coli* cells. We observe toxicity upon co-expression that was less pronounced as compared to e.g. ΔN-ClpC-F436A/ClpP co-expression (new Figure 6—figure supplement 1). Strong degradation of MecA upon ClpC co-expression demonstrates formation of an active MecA/ClpC/ClpP complex while potentially restricting toxicity at the same time (new Figure 6—figure supplement 1). This new finding substantiates our former observation that M-domain mutations cause adapter-independent ClpC activation. We suggest that MecA/ClpC exhibit restricted substrate specificity as compared to activated ClpC M-domain mutants, rationalizing an essential need for M-domain mediated activity control. In agreement with this model, we observed strong toxicity upon expression of *B. subtilis* ClpC-F436A but not ClpC wild-type in *B. subtilis* cells harboring the entire ensemble of ClpC interacting adapter proteins (Figure 6). This indicates that adapters convey an increased substrate specificity to ClpC preventing uncontrolled proteolysis upon ClpC activation – in contrast to ClpC M-domain mutants.

Major points:1) Flexible fitting was stated as being used to generate the atomic model of the ClpC-MecA structure, but very few details are given regarding its generation, aside from the fact that iModFit was used. Why was flexible fitting not used to generate a model of the ClpC double lockwasher, which is at higher resolution? The fitting shown in Figure 3 has much of the atomic model out of density, and should not be used for detailed structural interpretation. For example, in the second paragraph of the subsection “Head-to-head interactions of M-domains mediate formation of an inactive 194 ClpC resting state”, the authors state that the trans-acting arginine fingers are displaced away from the nucleotide binding pockets (Figure 3—figure supplement 1), but given the limited resolution of the map and poor fitting of the atomic model, this cannot be claimed. The reviewers agreed that it is fine to include side chains in the models depicted in Figure 3 and Figure 4, as these data are supported by biochemistry. However, when depositing atomic models based on the presented intermediate resolution EM, the models should only include the C-alpha's of amino acids.

Model generation and fitting are now clarified in the text in the Materials and methods part (subsection “Model building and fitting”). The *S. aureus* atomic model of ClpC and MecA monomers were generated using the server Phyre. They were modeled based on the *B. subtilis* ClpC atomic structures (PDB code 3pxi).No modeling was done de novo based on any of the EM structures, which are both at too low resolution for de novo modeling.Each protomer was then rigidly fitted manually using Chimera and refined using iModFit. For the ClpC resting state structure, real-space refinement in Phenix and Rosetta was performed trying to optimize as much as possible the geometry. Flexible, non-fittable loops where cut away from the final deposited pdb (PDBcode: 6EM9, 6EM8). Fittings were performed both on the C2 symmetrized and asymmetric maps and both maps and coordinates (only C-α) have been deposited (EMDB-3895, EMDB-3894). For the ClpC-MecA complex the *S. aureus* Phyre model was fitted into the map using Chimera and refined with iMODfit. The ClpC-MecA map and the fitted pdb (only C-alpha) have been deposited (EMD-3897, PDB ID 6EMW).

Following the suggestion of the reviewers, we submitted only C-α models of ClpC and ClpC-MecA as the maps are not at high-enough resolution to include information about side-chains.

We followed the criticism of the reviewers, removed Figure 3—figure supplement 1 and do no longer discuss positions of arginine fingers.

2) The ClpC-MecA structure is described as "asymmetric", but is the structure organized as a spiral, as has been shown in numerous other AAA ATPases? If not, this is novel and should be described in more detail.Also, the structures of Hsp104 and VAT in a steep lockwasher-like conformation were recently solved – are there any structural similarities with the resting state of ClpC?

The ClpC-MecA structure is at low resolution and the definition of asymmetry comes only from the fact that, similarly to the recently published Vps4 structures (Suns et al., 2017, EMD-6735) or VAT (Ripstein ZA et al., 2017, EMBD-8659), at high threshold, one subunit disappears before the others. The *S. aureus* ClpC-MecA structure is planar and overall is not different from the existing ClpC-MecA maps of *B. subtilis* (EMD-5607, EMD-5608, EMD-5610). The quality of this map is, however, better than the existing *B. subtilis* ClpC-MecA EM maps as the M-domains are clearly distinguishable. Apart from this, the map is at too low resolution to allow for meaningful comparison with other recently determined AAA ATPase structures of higher resolution. We therefore now omit commenting on asymmetry.

We have now compared all the lockwasher-like structures of AAA+ proteins by both overlapping the various helical models as well as comparing the rotation and shift angles required in each different helix to go from one protomer to the next. The following table summarizes the results:

**PDB code****ROTATION****SHIFT****HANDEDNESS**ClpC resting state53.53-14.99Left handed5vy8 (Hsp104-ADP)51.71-9.34Left handed5d4w (Hsp104 crystal)59.80-15.58Left handed5vya (Hsp104-ADP)60.486.25Right handed5kne (Hsp104-AMPPNP)52.50-8.86Left handed5ofo (ClpB-ATPgS + substrate)58.851.50Right handed

Structures overlapping and helical parameters show that the ClpC resting state is similar to some of the helical structures of Hsp104 obtained via X-ray crystallography and cryo EM. However, the helical axis of these assemblies have slightly different orientations (see Author response image 4), indicating that the protomer interfaces are slightly different from one another. It is currently not clear whether these differences are caused by sequence alterations between ClpC and Hsp104 or head-to-head MD contacts. The comparison with other helical assemblies of AAA+ proteins has been added to the revised manuscript.

**Author response image 4. respfig4:** Comparison of the helical assemblies of the ClpC resting state and the AAA+ disaggregase Hsp104. Images on the left show the full helical assembly and images on the right show two adjacent protomers with the helical axes.

3) It's puzzling that 3D classification was not performed at any point – this is regularly used before 3D refinement of a structure to identify a set of conformationally and compositionally homogeneous particles for processing, and then a classification without further alignment is performed after refinement to identify the subset of particles containing the highest resolution information. If these steps were performed and all 3D classes looked identical, this should be stated. Furthermore, it isn't clear which final density from Figure 3—figure supplement 1 was used for the structural analyses. The final structure from cryoSparc appears to be at higher resolution than the RELION structures – what was the reported resolution of this structure? Was C2 symmetry applied?

3D classification was actually performed but resulted in separation of only bad particles from the main population, rather than giving any information of conformational heterogeneity of the sample. Author response image 5 illustrates a chart of the 3D classification performed.

**Author response image 5. respfig5:** 3D classification of the ClpC dataset after cleaning via 2D classification.

The final density used was the C2 symmetrized map obtained from RELION (resolution 8.5Å). This map was originally chosen because the N-termini could be better defined. However, as pointed out by the reviewers, the C1 map from cryoSPARC appears to be better and it is probably more correct to interpret asymmetric maps. We have therefore prepared new main figures for the revised manuscript with the C1 structure obtained from cryoSPARC (8.4Å resolution).

Both maps are shown in new Figure 3—figure supplement 2, with their angular distribution and FSC curves. Also, both maps have been deposited.

4) The authors use glutaraldehyde crosslinking to analyze how MecA and the F436A mutation affect the formation of the decameric resting state of ClpC (Figure 5). Surprisingly, lane 9 shows that most of the double-Walker B mutant still forms the decamer (or even larger structure) in the presence of MecA. Was this trend also observed in the MecA-bound EM sample? In the Discussion, the authors state that "once formed, the ClpC6/MecA6 complex is stable and does not dissociate spontaneously". Why then does MecA-bound ClpC-DWB show any crosslinking larger than hexamers?

The reviewer is correct, the formation of high molecular weight products upon crosslinking of ClpC-DWB/MecA samples is unexpected. We speculate that a ClpC-DWB/MecA hexameric complex might be able to transiently interact with another complex via MecA-MecA interactions. This scenario is based on the finding that MecA can target itself for degradation and supported by the observation that at early time points of crosslinking (2 min) mostly hexameric complexes are formed and larger crosslink products are only detected at the later time point (10 min). We assume that such ClpC/MecA complex interactions are transient and therefore not detectable by size exclusion chromatography while being frozen upon chemical crosslinking. This clarification has been added to the revised manuscript.

In the EM sample no higher-order oligomers were observed.

5) It is surprising that, based on gel filtration results (Figure 5), the formation of ClpC decamers is ATP-dependent, whereas hexamer formation is not, and the authors should try to discuss this. Furthermore, how does the limited interaction surface of MDs (~ 50 A2) compare to the AAA interfaces? Is this interaction indeed substantial and strong enough to disrupt the interactions in a planar, ATP-bound AAA ring when MecA is absent?

The reviewers are not correct, resting state (ClpC wild-type) and hexamer

(ClpC-F436A and ClpC wild-type + MecA) formation as analyzed by SEC are both ATP-dependent (Figure 5). The trailing of ClpC wild-type observed in absence of ATP does not reflect hexamer formation but assembly into larger, unstable intermediate oligomeric forms. In presence of glutaraldehyde crosslinker these oligomers are crosslinked to a high molecular weight complex (Figure 5).

As discussed in the points above, multiple helical arrangements of AAA+ proteins have been recently described and the ClpC resting state conformation relates to some of them. Adjacent AAA domains share large surface interactions both in an helical as well in a planar conformation. Therefore the mechanisms of activation of these proteins must be regulated by detailed reorganization of the ATP pockets and cannot just simply be described in terms of surface of interactions. We are missing a high-resolution structure of the activated ClpC-F436A MD mutant. Therefore, it is currently not possible to say how the MDs are influencing AAA domain interactions and nucleotide binding. This is a central point and we hope that higher-resolution structures of ClpC and ClpC-F436A where details about ATP binding sites can be described will be able to shed light on this point.

6) MecA has been shown in B. subtilis to promote assembly from a lower assembly state (monomeric/dimeric) to the active hexamer. Although the authors convincingly show the existence of an inactive higher order assembly of S. aureus ClpC, it remains unclear why such a complex is beneficial. This should be discussed.The authors suggest that activation upon MecA association occurs via monomeric ClpC. Can they provide evidence for that? Can it be excluded that the double-spiral dissociates into two single spirals that then transition more directly into the hexameric state?

We suggest that resting state formation can prevent inactive ClpC subunits from (non)specific interactions with other cellular components or protect ClpC from degradation by other proteases as compared to freely accessible ClpC monomers. This point has been added to the Discussion section. We would like to stress that the observation of severe toxicity of *B. subtilis* ClpC-F436A cannot be explained by the former regulatory model involving only transitions between monomeric and hexameric ClpC but by our newly derived model involving resting state formation.

We assume that ClpC activation relies on MecA binding upon dissociation of ClpC subunits from the resting state. Peripheral ClpC subunits do not show complete densities in the cryo-EM model, suggesting increased flexibility and dynamics. Resting state formation precludes MecA binding as M- and N-domains are rendered inaccessible for MecA interaction. Therefore dissociation of ClpC subunits seems prerequisite for subsequent MecA interaction. We however cannot exclude that double-spiral dissociation takes place and allows for immediate transition into the hexameric state upon MecA binding. We have clarified this aspect in the revised manuscript (Discussion, fifth paragraph).

7) It is proposed that the MecA adaptor is degraded and ClpC consequently inactivated when substrates are no longer available. However, the strong cytotoxicity of MD mutants in E. coli would suggest that the substrate specificity of ClpC/P is rather broad. The authors also speculate that deregulated ClpC/P may go after newly synthesized proteins, which raises the question whether ClpC/P would indeed ever run out of substrates to then be inactivated. Is the assumption that MecA makes ClpC/P more specific and less promiscuous than MD mutants like F436A? The authors should try to address this in their Discussion.

Our analysis indicates that ClpC-activating adapter proteins exhibit a higher degree of substrate specificity as compared to constitutively active ClpC M-domain mutants. Co-expression of ClpC/ClpP/MecA is less toxic as compared to ClpC-F436A/ClpP (new Figure 6—figure supplement 1/D). Also, *B. subtilis* ClpC-F436A is highly toxic whereas the same levels of ClpC wild-type do not exhibit any detrimental effect on growth of *B. subtilis* cells harboring the complete set of ClpC adapter proteins (Figure 6). This difference in substrate selection between adapter-activated ClpC and activated ClpC M-domain mutants has been stressed in the revised manuscript (end of the Discussion section).